# Mathematical Capabilities of ChatGPT

**Simon Frieder**[*,1]**, Luca Pinchetti**[1]**, Alexis Chevalier**[3]**, Ryan-Rhys Griffiths**[4]**,
Tommaso Salvatori**[2,8]**, Thomas Lukasiewicz**[8,1]**, Philipp Petersen**[6,7]**, Julius Berner**[5]

[1]Department of Computer Science, University of Oxford, Oxford, UK
[2]VERSES AI Research Lab, Los Angeles, US
[3]School of Mathematics, Institute for Advanced Study, Princeton, US
[4]Department of Physics, University of Cambridge, Cambridge, UK
[5]Department of Computing and Mathematical Sciences, Caltech, Pasadena, US
[6]Faculty of Mathematics, University of Vienna, Vienna, Austria
[7]Research Network Data Science, University of Vienna, Vienna, Austria
[8]Institute of Logic and Computation, Vienna University of Technology, Vienna, Austria
https://ghosts.friederr.org

## Abstract

We investigate the mathematical capabilities of two versions of ChatGPT (released 9-January-2023 and 30-January-2023) and of GPT-4 by testing them on publicly available datasets, as well as hand-crafted ones, using a novel evaluation scheme. In contrast to formal mathematics, where large databases of formal proofs are available (e.g., mathlib, the Lean Mathematical Library), current datasets of natural-language mathematics used to benchmark language models either cover only elementary mathematics or are very small. We address this by publicly releasing two new datasets: GHOSTS and miniGHOSTS. These are the first natural-language datasets curated by working researchers in mathematics that (1) aim to cover graduate-level mathematics, (2) provide a holistic overview of the mathematical capabilities of language models, and (3) distinguish multiple dimensions of mathematical reasoning. These datasets test, by using 1636 human expert evaluations, whether ChatGPT and GPT-4 can be helpful assistants to professional mathematicians by emulating use cases that arise in the daily professional activities of mathematicians. We benchmark the models on a range of fine-grained performance metrics. For advanced mathematics, this is the most detailed evaluation effort to date. We find that ChatGPT and GPT-4 can be used most successfully as mathematical assistants for querying facts, acting as mathematical search engines and knowledge base interfaces. GPT-4 can additionally be used for undergraduate-level mathematics but fails on graduate-level difficulty. Contrary to many positive reports in the media about GPT-4 and ChatGPT's exam-solving abilities (a potential case of selection bias), their overall mathematical performance is well below the level of a graduate student. Hence, if you aim to use ChatGPT to pass a graduate-level math exam, you would be better off copying from your average peer!

---

[*]Corresponding author: simon.frieder@cs.ox.ac.uk.

37th Conference on Neural Information Processing Systems (NeurIPS 2023) Track on Datasets and Benchmarks.

# 1  Introduction

Since its release in November 2022, the language model *Chat Generative Pre-trained Transformer* (ChatGPT) has rapidly become a widely known question-and-answer dialogue system. ChatGPT has been referenced in mainstream media across the globe [1–4] and across all major internet platforms [5, 6]. With similar reactions, the release of ChatGPT's successor, GPT-4, followed in March 2023 [7].

The performance of ChatGPT has been analyzed in a large number of exam-related use cases, with varying degrees of scientific rigor, ranging from detailed studies to anecdotal evidence. Use cases include passing the *United States Medical Licensing Examination* (USMLE) [8], scoring highly on the *Psychology Today* Verbal-Linguistic Intelligence IQ Test [9], and answering (and generating) Operations Management exam questions that were deemed to be within the scope of a typical MBA curriculum [10], all with a performance that elicited a positive sense of surprise from the authors. In turn, the performance of GPT-4 even surpasses that of ChatGPT on a large batch of academic and professional exams [7, Table 1]. Such strong task-related performance indicates that large language models (LLMs) could be frequently used as assistants in many domains.

In this paper, we introduce a new dataset called GHOSTS, which measures the advanced mathematical abilities of LLMs. Using this dataset, we perform a detailed analysis of the mathematical capabilities of ChatGPT on two of its versions, the 9-January-2023 and the 30-January-2023 version. Note that, according to the release notes, the 30-January-2023 version should possess "*improved factuality and mathematical capabilities*" [11]. We further examine the performance of GPT-4 on a smaller dataset called miniGHOSTS, which exhibits statistics similar to the larger GHOSTS dataset. We further make available microGHOSTS, which itself is a subset of miniGHOSTS, and is designed to facilitate pre-screen of language models, by incurring minimal human evaluation costs. Our analysis includes but is not limited to testing how many of the skills necessary to do professional mathematics can be emulated by these models. Examples of such skills are the ability to answer computational questions, the ability to complete mathematical proofs that have gaps or missing steps, the ability to solve questions that are more focused on deep insights and original solutions, such as those of mathematical olympiads, and the ability to survey the literature and think across domains. None of the previous benchmarks (see Section 2) cover such a broad range of mathematical abilities.

To achieve the goals outlined above, GHOSTS consists of carefully composed prompts aimed at testing different aspects of LLMs related to mathematical comprehension; see Section 3. This includes both hand-crafted prompts as well as samples from existing datasets that were devised to test models specifically trained for mathematical comprehension [12, 13].

For brevity, we use the expression "**(Chat)GPT**" to refer collectively to both the ChatGPT and GPT-4 language models. We refer to Appendix C for further details regarding different (Chat)GPT versions.

To evaluate the output of (Chat)GPT, we designed a thorough testing methodology, including warning and error codes that represent various possible failure modes of (Chat)GPT. We score (Chat)GPT's responses, report on the results using this methodology, and compare (Chat)GPT to a selection of state-of-the-art models trained for mathematical comprehension. In summary, the contributions of this article are threefold:

- **Benchmark for testing the mathematical capabilities of LLMs:** We introduce a new natural-language mathematics dataset, called GHOSTS[2], to test the capabilities of LLMs across a range of aspects regarding advanced mathematical comprehension; see Section 3. It consists of two subdatasets derived from state-of-the-art datasets of mathematical queries for language models. Additionally, we devise four hand-crafted subdatasets covering further mathematical tasks. Parts of our dataset consist of problems that were selected to have a high probability of not being in the data on which (Chat)GPT was trained; see tags D1-D3 from Table 1.

---

[2]The GHOSTS dataset is available at `github.com/friederrr/GHOSTS`, and the project is hosted at `ghosts.friederrr.org`.

- **Insight for mathematical use of (Chat)GPT:** Based on our benchmark, we show for which types of questions and which domains of mathematics, (Chat)GPT may be useful and how it could be integrated into the workflow of a mathematician. On the other hand, we identify the failure modes, as well as the limits of its capabilities. This can aid future efforts to develop LLMs that perform better in mathematics. Our analysis is akin to a *mathematical model card* [14] in terms of intended use, metrics, evaluation data, and qualitative analyses, as mathematical strengths and weaknesses of (Chat)GPT are summarized; see Section 4.

- **Evaluation of improvements of (Chat)GPT:** We can further use our benchmark to track the mathematical capabilities of (Chat)GPT variants over time. As a first step, we analyze the impact of the upgrade from the 9-January-2023 to the 30-January-2023 version of ChatGPT, which promises a better mathematical performance according to the release notes. Then, we proceed to investigate what performance increases the successor GPT-4 brings; see Section 4.1.

## 2  Related Work

As a language model, (Chat)GPT can be universally employed to perform mathematical reasoning and, therefore, has to compete with technologies in this space that are sometimes decades old. Performing mathematical reasoning in an automated way has a long history and can be traced back to 1959 [15], the most focus being devoted to proving theorems [16]. According to Harrison [17], there is a realization that classical approaches, using a symbolic encoding of mathematics, have reached a "plateau".

On the other hand, there is now a growing body of literature on learning mathematical relationships directly in a supervised-learning manner [18–20] or by using LLMs to perform mathematical reasoning directly on mathematics encoded in natural language [21]. Sometimes, the distinction is blurred, because architectures of LLMs can also be used in a supervised-learning setting and have been employed successfully in learning mathematical relationships, such as between the syntactical form of a function and its integral [13, 22].

Among the supervised approaches, we mention [13], where a Transformer architecture [23] was used to generate symbolic, closed-form solutions to integrals and first and second-order differential equations, which outperformed classical solvers[3], such as Mathematica, MATLAB, and Maple by at least 14% on a test set of integration problems. On the task of solving differential equations, the Transformer-based approach still exceeds the classical approach, but by a smaller margin (at least 4% in the case of first-order differential equations and with more varied results for second-order equations).

Regarding LLMs, recent ones, for instance, PaLM [24] (released in 2022), are tested only on elementary-level mathematical reasoning datasets, such as the MathQA or GSM8K datasets [25, 26]. We hypothesize that this is due to a lack of advanced-level natural language mathematics datasets. Moreover, the results obtained indicate that the models at that time had difficulty with much simpler datasets than ours. For example, the version of PaLM with 540 billion parameters only correctly solves 58% of the problems of the GSM8K dataset, even with chain-of-thought prompting and access to an external calculator [24, Table 10]. This model nonetheless outperforms GPT-3 [27], which only achieves 54% on the same dataset. Variations of BERT [28] have been shown to only solve between 28% and 37% of the problems when fine-tuned and tested on the *Algebra Question Answering with Rationales* (AQuA-RAT) dataset [29], which is the direct predecessor of MathQA. For some models, such as BLOOM [30] or the LaMDA model [31] (both released in 2022), an evaluation of the mathematical reasoning capability is entirely missing. An up-to-date survey on mathematical datasets and the performance of various LLMs can be found in [32].

Most similar to our dataset is the NATURALPROOFS dataset [33] and the NATURALPROOFS-GEN dataset [34]. In this paragraph, we illustrate the similarities and differences between these datasets and ours. NATURALPROOFS and NATURALPROOFS-GEN are similar among

---

[3]For a given prompt, the computer algebra system is considered to have failed if it does not provide a closed-form solution or times out after 30 seconds (in case of Mathematica).

themselves and cover graduate-level mathematics by focusing on data from ProofWiki[4] (the latter dataset), as well as on the Stacks Project[5] and two open-source textbooks (the former dataset). Using the LaTeX source code, which is available for all these resources, annotated theorems and their proof graphs are extracted. The annotations consist of reference graphs highlighting references to other theorems or definitions, the idea being that these references capture the "skeleton" of a proof. This task resembles the mathematical abilities that the *Named Theorem Proof Completion* subdataset from the GHOSTS dataset evaluates (see Table 1), although (1) we only retrieve a single reference, and (2) (Chat)GPT, as far as known, does not use training objectives that make use of information from data annotation, in contrast to models evaluated in [33, 34].

Our framework pertains to general language model evaluation, which may be presented in a black-box manner (as is the case for (Chat)GPT), and therefore does not allow to leverage any additional information, such as reference graphs. This is also reflected in the human evaluation schema introduced in [34, Table 24], which classifies common model mistakes. As reference graphs form the foundation of how the mathematical proofs are engineered, many elements of the evaluation schema are strongly tailored toward this representation of mathematical data. Our benchmark is not reference-centric and therefore allows evaluations of *any* type of proof (including computations, as featured in the *Symbolic-Integration* subdataset, which we consider to be a particular kind of proof). Therefore, our methodology includes further and more general failure modes to make for a more fine-grained evaluation that explains the nature of the errors. We refer to Appendix A for further related works.

## 3   The GHOSTS, miniGHOSTS, and microGHOSTS Dataset

We assess the mathematical reasoning capabilities of two ChatGPT versions, 9-January-2023 and 30-January-2023, and of GPT-4 by first creating a collection of 709 prompts from various sources, and subsequently evaluating the models on (subsets of) these data points. We rate the corresponding outputs provided by the models and collect statistics, such as error types, output lengths, or the stability of the answer under prompt engineering, see Sections 3.2 and 4 and Appendices B and D. This yields a total of 1636 ratings by human experts.

We divide our dataset, the entire collection of prompts, into six *subdatasets*, called

- $\boldsymbol{G}$*rad-Text*,
- $\boldsymbol{H}$*oles-in-Proofs*,
- $\boldsymbol{O}$*lympiad-Problem-Solving*,
- $\boldsymbol{S}$*ymbolic-Integration*,
- $MA\boldsymbol{T}H$,
- $\boldsymbol{S}$*earch-Engine-Aspects*,

which, in turn, consists of multiple *files*, see Table 1. The boldface letters make up the **GHOSTS** acronym. Details on motivation, composition, collection process, and intended uses of the GHOSTS dataset are summarized in our datasheet in Appendix H, Sections H.1, H.2, H.3, and H.5, respectively.

GPT-4 was evaluated on a subset of 170 prompts, which we call the **miniGHOSTS** dataset. Specifically, after having created the GHOSTS dataset, we heuristically selected a subset of 10 prompts from each file of the subdatasets included in GHOSTS, having the same mean rating and the same standard deviation (of ChatGPT's output) as the original file; see also our datasheet in Appendix H for more information. In this sense, these subsets can be considered to have the most relevance by capturing the "essence" of the model performance in the respective file. The role of miniGHOSTS is thus to reduce the costly evaluation procedure of the full GHOSTS dataset on a new language model. The **microGHOSTS** dataset represents a further reduction of the miniGHOSTS dataset, where one question was extracted from each file making up the miniGHOSTS dataset, for a total of 14 questions. The role of microGHOSTS is to allow rapid pre-screening of a language model, where each of the 14 questions was chosen to be a question representative of the mathematical problems in that file, as well as being a problem that (Chat)GPT typically struggled with. Reference solutions, explanations, and known LLM failure modes are provided and discussed for the microGHOSTS dataset in order to aid raters who are not mathematically trained. For more information, see Appendix E.

---

[4] https://proofwiki.org/
[5] https://github.com/stacks/stacks-project

Table 1: A summary of all the files from all the subdatasets comprising our GHOSTS dataset, together with their size, i.e., the number of prompts and their associated attribute tags. The tags M*i*, Q*i*, and D*i* relate to the level of Mathematical difficulty, the Question type, and the Out-of-Distribution type from Section 3.1, respectively. For the *Olympiad-Problem-Solving* subdataset, 24 further prompts were created where prompt engineering was applied, see Appendix 4.2. These 24 prompts do not count towards the 709 total prompts, only towards the 1636 evaluations.

| Subdataset Name | Size | Comprised of the `json` file(s) | Tags |
|---|---|---|---|
| ***G****rad-Text* | 28 | W. Rudin, Functional Analysis (ch. 1) | M3 Q4 |
| | 15 | W. Rudin, Functional Analysis (ch. 2) | M3 Q4 |
| | 37 | J. Munkres, Topology (ch. 1) | M3 Q4 |
| | 29 | J. Munkres, Topology (ch. 2) | M3 Q4 |
| | 21 | R. Durrett, Probability Theory | M3 Q4 |
| ***H****oles-in-Proofs* | 60 | Proofs Collection A | M3 Q1 Q2 Q5 |
| | 52 | Proofs Collection B Prealgebra | M1 Q5 |
| | 50 | Proofs Collection B Precalculus | M1 Q5 |
| ***O****lympiad-Problem-Solving* | 101+24 | Olympiad Problem Solving | M4 Q4 D2 |
| ***S****ymbolic-Integration* | 100 | Symbolic Integration | M2 Q3 D1 |
| *MA**T**H* | 50 | MATH Algebra | M1 M2 M3 Q3 Q4 |
| | 50 | MATH Counting and Probability | M1 M2 M3 Q3 Q4 |
| | 18 | MATH Prealgebra | M1 Q3 Q4 |
| | 20 | MATH Precalculus | M1 Q3 Q4 |
| ***S****earch-Engine-Aspects* | 30 | Definition Retrieval | M3 Q2 D3 |
| | 30 | Reverse Definition Retrieval | M3 Q1 Q2 D3 |
| | 18 | Named Theorem Proof Completion | M3 Q2 Q5 D3 |

## 3.1 Subdatasets

The subdatasets that make up our GHOSTS dataset are summarized in Table 1. In the following, we describe each subdataset in more detail.

**Grad-Text.** This subdataset consists of a collection of books (R. Durrett's *Probability Theory* [35], J. R. Munkres *Topology* [36] and W. Rudin's *Functional Analysis* [37]) that are widely used in universities to teach upper undergraduate or first-year graduate courses in a degree in mathematics. We have used most of the exercises from these books' first and second chapters, except for [35], where we only used exercises from the first chapter, which was longer than the other books' chapters.

**Holes-in-Proofs.** This subdataset consists of a number of proofs sourced from `math.stackexchange.com`, as well as some proofs sourced from books (S. Axler's *Linear Algebra Done Right* [38] and W. Rudin's *Principles of Mathematical Analysis* [39]), and from the MATH dataset [12], where parts of the proofs were intentionally deleted and the LLM was prompted to fill in the gaps: This was done either by (1) using a `MISSING` token, (2) finishing the proof early and prompting the LLM to complete it, or (3) explicitly asking for certain conditions or results.

**Olympiad-Problem-Solving.** This subdataset consists of a selection of exercises from A. Engel's *Problem-Solving Strategies* [40] book, which is often used to prepare for mathematical competitions. We selected and graded the LLM outputs on one hundred exercises drawn from all chapters.

**Symbolic-Integration.** This subdataset consists of random samples of integrals from the test set of [13]. There are three ways in which integrals are generated in [13]: *Forward generation* (FWD), *Backward generation* (BWD), and *Backward generation with integration by parts* (IBP). We sample 21 integrals from FWD test set, 20 integrals from the BWD test set, and 59 integrals from the IBP test set. As these integrals are given in Polish/prefix notation, a natural-language prompt conversion of them is unlikely to be witnessed in the training dataset of (Chat)GPT. The assessment was done by verifying the correctness of the output both by using Mathematica, as well as making use of the provided solutions

(in Polish notation), which [13] generated using SymPy. In particular, we notice that all integrals in this dataset have solutions that can be expressed using elementary functions.

**MATH.** This subdataset consists of a random sample of problems from the MATH dataset [12]. The latter dataset attaches a level of difficulty to each problem. We focused on two domains, Algebra and Probability Theory, and sampled an equal number of problems at each level of difficulty.

**Search-Engine-Aspects.** This subdataset consists of problems that were not sampled from a particular source but generated by a human expert in the field. In the file *Named Theorem Proof Completion*, we focused on prompting the LLM to provide proof outlines of various theorems that are sufficiently well-known within Functional Analysis to have names. In the *Definition Retrieval* file, we asked the LLM to correctly state various definitions centered around Functional Analysis and Topology. In contrast, in the *Reverse Definition Retrieval* file, we verified whether the LLM was able to deduce the name of a mathematical object by describing its properties.

Because input to (Chat)GPT is purely textual (at the time of writing), certain types of questions that might be stated and solved in a non-text-based fashion (e.g., questions involving graphical diagrams, without text explaining the diagram[6], as occasionally occur in [40]), have been excluded. Our subdatasets can be categorized along the following dimensions (see Appendix B.1 for more details):

- **Mathematical difficulty (ascending):** (M1) Elementary arithmetic problems, (M2) Symbolic problems, (M3) (Under)graduate-level exercises, (M4) Mathematical olympiad problems.

- **Question type:** (Q1) Stating mathematical facts, (Q2) Overview-type review questions, (Q3) Computational questions, (Q4) Theorem proofs or puzzle solutions, (Q5) Proof-completion questions.

- **Types of high out-of-distribution likelihood:** (D1) Nontrivial problem encoding, (D2) Succinct solution, (D3) Spoken dialogue.

The existing datasets of natural-language mathematics are far from covering all possible combinations across these dimensions. In our well-crafted GHOSTS datasets, we have striven to cover each of these aspects individually, as can be seen in Table 1. The next section specifies the format of our dataset and the methodology for analyzing (Chat)GPT's output.

### 3.2 Format

The format of each of the subdatasets that make up our GHOSTS dataset follows the same convention. Each subdataset consists of JSON-formatted files, and our format is similar to, e.g., the AQuA-RAT dataset [29]. A single data point[7] in a file has the following form:

```
"prompt": "Let $X$ be a topological vector space. All sets mentioned
    below are understood to be the subsets of $X$. Prove the
    following statement: If $A$ and $B$ are compact, so is $A + B$",
"output": "The statement is wrong in general. Consider the example $A
    = [-1,1] \times {0}$ and $B = {0} \times [-1,1]$. Then $A$ and
    $B$ are compact but $A + B = [-1,1] \times [-1,1]$ is not
    compact."
"rating": "2",
"errorcodes": ["e3", "e5_2", "e5_4"],
"warningcodes": [],
"comment": "The given $A + B$ actually *is* compact.",
"msc": ["46A03"],
"ref": "Rudin-Functional Analysis-Second-Ed.Part1-ex3/d-page38",
```

---

[6]See, e.g., Exercise 15 in [40, Chapter 2], which asked the reader to inspect a figure on which the problem is based.

[7]The JSON object of an output of the 30-January-2023 version of ChatGPT, as identifiable by the timestamp at which the output was generated, is shown. The prompt comes from the "W. Rudin, Functional Analysis (ch. 1)" file from the *Grad-Text* subdataset.

```
"confidence": "high",
"timestamp": "2023-01-31"
```

We require each data point to have the same JSON keys as in this example, some of which may be empty depending on the prompt. Among the listed keys, the `rating` key stands out as the most fundamental one. Its value serves as a condensed representation of the mathematical capability of the tested language model, compressed into a one-dimensional measure ranging from `1` (lowest) to `5` (highest). A more nuanced and fine-grained perspective on the mathematical capabilities is provided by the `errorcodes` and `warningcodes` keys. The `msc` key denotes the *mathematics subject classification*. We explain each JSON key in Appendix B.2. For end-users of (Chat)GPT, it is desirable to avoid having a long-winded dialogue to arrive at a solution. Therefore, we require that (Chat)GPT provides us with the correct solution given only the input prompt without any subsequent interaction.

### 3.3 Human Input in Dataset Creation and Mathematical Evaluation

For all data points, the values of the keys `rating`, `errorcodes`, `warningcodes`, `comment`, and `confidence` were manually labeled, without any automation. The `msc`, `ref`, and `timestamp` keys were populated in a semi-automatic way, since their values change only slightly within the same subdataset.

Two of the subdatasets, the *MATH* subdataset and the *Symbolic-Integration* subdataset, use prompts taken from existing datasets, the MATH dataset by [12] and the dataset comprising integrals from [13], respectively. This was done to compare how (Chat)GPT performs against existing state-of-the-art models that use these datasets, see Section 4. Nonetheless, significant additional annotation effort was involved, since, in both cases, the authors rated the output. Furthermore, in the second case, the data are publicly presented in a Polish notation format, and conversion was necessary. The prompts of the other subdatasets were hand-crafted by the authors. We refer to Appendix B.6 for more information on different aspects of human effort in dataset creation.

## 4 Results

Will ChatGPT get you through a university math class? No, you would be better off copying from your average peer—unless it is undergraduate mathematics, for which GPT-4 can offer sufficient (but not perfect) performance.

If we take a rating of `3.5`, the average between the lowest and highest rating, to be the threshold between success and failure, then Figure 1 shows that for a majority of subdatasets, both versions of ChatGPT will not pass. However, for GPT-4, the situation is different, and, on miniGHOSTS, it passes (sometimes barely) on all subdatasets files, except W. Rudin, Functional Analysis (ch. 2), which tests graduate-level mathematical knowledge and the Olympiad Problem Solving file, which tests mathematical problem-solving skills. We note that, unless otherwise stated, we do not use prompt-engineered questions in the results presented here (see Appendix 4.2).

We first focus on the results of the 9-January-2023 version of ChatGPT and note that the results for the 30-January-2023 are very similar, as can be inferred from the figures. On average, the 9-January-2023 version achieves a rating of `3.20` with a standard deviation[8] of `1.23`. It performs particularly poorly on proof-based questions in the style of graduate-level exercises or mathematical olympiads, as well as more complicated symbolic calculations. We note that prompt engineering only slightly improved the results for such complex questions; see Appendix 4.2. However, in tasks that only required filling in gaps or stating mathematical facts, ChatGPT was mostly able to achieve a score above `3.5`. In particular, ChatGPT was strong at recognizing the context of the question, and the notation of the output almost always matched the one given in the prompt, see Figure 4 in the appendix. Generally, Figure 1 indicates that the ratings closely correspond to how mathematicians would rank the difficulty of the exercises. In this context, we note that the length of the prompt does not have a clear effect on the rating; see Figure 9 in the appendix. We present results for different mathematical fields in Figure 5 in the appendix. For a detailed qualitative analysis

---

[8]We use Bessel's correction term to obtain an unbiased estimate of the variance.

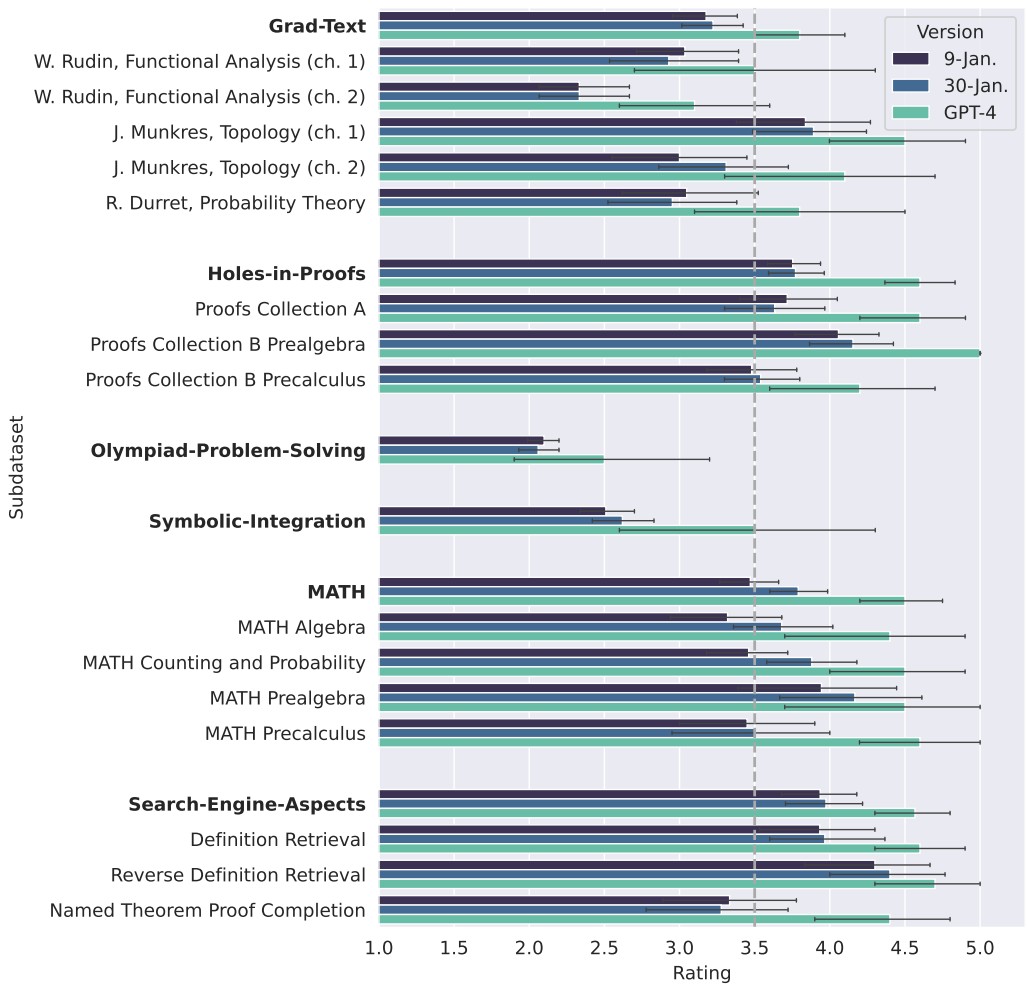

Figure 1: Average rating for each file in each subdataset (bold) of GHOSTS on the 9-January-2023 and the 30-January-2023 versions of ChatGPT and for miniGHOSTS on GPT-4. Note that the maximal ranking is 5 and the minimal ranking, where the question was at least understood, is 2, see Appendix B.4; the lower rating of 1 indicates that the answer completely misses the question. Thus, a reasonable passing grade, i.e., 50% of points, corresponds to a score of 3.5, as indicated by the vertical dotted line. The error bars represent 95% confidence intervals.

of the results on the different subdatasets, we refer to Appendix D.1. Finally, we note that (Chat)GPT almost never expressed any form of uncertainty, even if its output has been completely wrong; see Appendix D.2.

Comparing ChatGPT to the performance obtained by [13], who correctly solved nearly 100% of the integrals in a collection of 500 test equations [13, Table 3], the 9-January-2023 version of ChatGPT achieves an average rating of 2.51 (standard deviation: 0.87) on our random sample of their dataset (after conversion from Polish notation to LaTeX). Specifically, a rating of 2 is dominating 70% of the time, followed by a rating of 3 and 4 for 13% of the prompts each; see also Figure 7 in the appendix. GPT-4 achieves an average of 3.50 (standard deviation: 1.43), barely a passing grade, on the corresponding subset from miniGHOSTS. These scores trail far behind the performance achieved by the model in [13]. The situation is similar when comparing ChatGPT to Minerva [21, Table 3]. Their best model achieved an accuracy of 50% on the MATH dataset [12]. However, the 9-January-2023 version of ChatGPT achieves a perfect score only on 29% of our random samples from the MATH

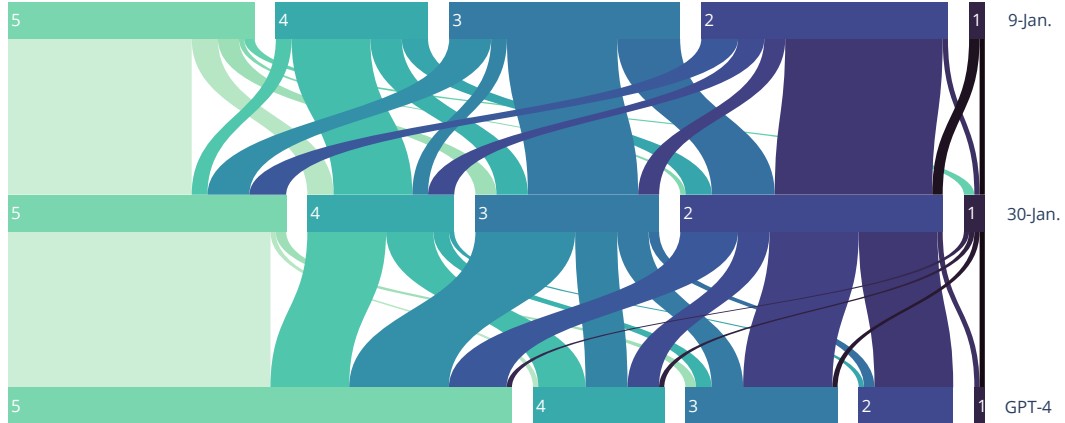

Figure 2: A Sankey diagram of how the ratings evolve from 9-January-2023 ChatGPT to 30-January-2023 ChatGPT to GPT-4 (from top to bottom), with all models evaluated on miniGHOST. While grades on the 9-January and 30-January models are shuffled between the ChatGPT versions, the overall performance remains approximately the same. However, we observe a significant increase in perfect ratings, i.e., a score of 5, for GPT-4.

dataset (which is above the total average of 25% of data points across all subdatasets in which this version achieves a perfect score), see Figures 6 and 7 in the appendix. In contrast, GPT-4 performs substantially better and obtains a score of 5 on 70% of the corresponding questions within the miniGHOSTS dataset, see Figure 7 in the appendix.

## 4.1 Quantitative Comparison of (Chat)GPT Versions

The ensuing model version, 30-January-2023, overall performed similarly with an average rating of 3.29 (standard deviation: 1.28), although performance was inconsistent across subdatasets and on some subdatasets marginally better, see Figure 1. A significant jump in performance could only be observed for GPT-4, which achieved a substantially higher average rating of 4.15 (standard deviation: 1.12). We note that the evaluation of GPT-4 is only on the miniGHOSTS dataset, i.e., a subset of GHOSTS. Nonetheless, these findings send a clear message that the performance of GPT-4 dominates the performance of ChatGPT (both versions), see Figure 1.

Figure 2 shows how the ratings change between the different versions of (Chat)GPT. Surprisingly, one can see a shuffling of the grades for the two ChatGPT versions, even though the counts in each grade bracket stay approximately the same. For instance, there are roughly the same amount of outputs that received grade 4, yet less than half of the prompts were the same between model changes. Appendix D.4 provides different perspectives on this and reinforces the mixed performance increase that the 30-January-2023 model brings. For GPT-4, we see that the percentage of perfect ratings almost doubles, while the percentage of prompts that are not understood or completely wrong (i.e., ratings of 1 or 2) approximately halves as compared to the ChatGPT versions.

Analysis of (Chat)GPT's output and our warning codes reveal that GPT-4 provides even longer ("rambling") answers, whereas ChatGPT usually answered the question without giving any additional context about the topic, see Figures 6 and 8 in the appendix. The answer style of GPT-4 was often beneficial (resulting in better overall scores) but sometimes reduced the readability of the output. Furthermore, we found the behavior of GPT-4, compared to ChatGPT, to be more opinionated. Finally, despite its better overall performance, GPT-4 still seems to be vulnerable to mistakes in seemingly simple calculations. We refer the reader to Appendix D for further results on the models' performance.

## 4.2 Prompt Engineering

One interesting finding of our study is related to performing prompt engineering on mathematical questions. Prompt engineering was solely carried out on questions from the *Olympiad-Problem-Solving* subdataset, and prompt-engineered questions consist of lists consisting of

two JSON objects. These lists contain the original question that was not prompt-engineered, as well as the prompt-engineered question. The latter question is identified, as it contains the string `<prompt engineered>` as the value in the `comment` key. These lists containing prompt-engineered questions are in the same hierarchy in the JSON file as the other questions from the subdataset.

About 20% of the questions were prompt-engineered: In these cases, ChatGPT was instructed to proceed either step-by-step, by prefixing the sentence "Let's answer this question step by step." or by adding words that formulate the mathematical task in a more explicit way, i.e., by adding "Prove that..." or "Show that..." to the prompt[9]. Instructing ChatGPT to proceed step-by-step in this way was shown to increase the performance of GPT-3 on datasets that test mathematical reasoning (e.g., GSM8K); furthermore, this is a type of prompt engineering that is recommended by OpenAI in their *cookbook* to improve reliability[10].

As a result of prompt engineering, for the 9-January-2023 version of ChatGPT, the number of wrong statements and computations (i.e., error codes `e2`, `e3`, and `e4`) decreased, while the number of errors rooted in faulty logic (i.e., error code `e5`) actually increased. Overall, prompt engineering improves the average rating only slightly, see Figure 3 from the Appendix.

For the questions from *Olympiad-Problem-Solving* that were selected for the miniGHOSTS dataset, we allow to sample from the entire *Olympiad-Problem-Solving* subdataset, since the goal of miniGHOSTS is not to measure prompt-engineering effects. Therefore, some of the questions in the miniGHOSTS version of the *Olympiad-Problem-Solving* subdataset contain prompt-engineered questions. The `<prompt engineered>` string was therefore removed from the comments in the miniGHOSTS dataset.

## 5 Conclusion

We have examined the behavior of (Chat)GPT across various tasks that test different aspects of mathematical skill. Contrary to the media sensation that (Chat)GPT has caused, (Chat)GPT is not yet ready to deliver high-quality proofs or calculations *consistently*. At the same time, the quality of the answers can be positively surprising. Moreover, our evaluation of GPT-4 on the miniGHOSTS dataset reveals promising improvements over ChatGPT's performance. In Appendix G, we collect the best and worst results for a number of selected subdatasets. The best responses can be seen to justify the media sensation. It thus seems fair to say that (Chat)GPT is *inconsistently bad* at advanced mathematics: While its capabilities generally drop with the mathematical difficulty of a prompt, it does give insightful proofs in a few cases.

However, (Chat)GPT falls short of achieving the same performance as models specifically trained for single tasks. These models, in contrast, lack the flexibility of (Chat)GPT, which is a *universal* tool suitable for any area of mathematics. In fact, (Chat)GPT's ability to search for mathematical objects, given information about them, is where it shines. For a user that is already sufficiently mathematically proficient to discern the correctness of (Chat)GPT's output, (Chat)GPT can be integrated as an assistant in the user's workflow. It can function as a search engine or knowledge base to speed up various lookup tasks, as they often occur at certain stages of mathematical research.

Due to the prohibitive annotation effort, the GHOSTS dataset is not yet large enough to significantly improve the mathematical capabilities of LLMs by fine-tuning them on GHOSTS; though we believe it is sufficiently comprehensive to allow an evaluation and comparison of LLMs (and more rapid evaluation using miniGHOSTS and microGHOSTS, respectively). We encourage other researchers to mine our dataset beyond the descriptive statistics that we computed to gain a deeper understanding of how LLMs behave on mathematical tasks. Finally, we hope that our work motivates other mathematicians to contribute to this growing field, by evaluating their LLMs on micro/mini/GHOSTS in order to establish a thorough benchmark for assessing the mathematical abilities of LLMs.

---

[9]Some prompts (e.g., the ones taken from the book by Engel [40]) only contain a mathematical statement, without a clear instruction; for example, "*An $a \times b$ rectangle can be covered by $1 \times n$ rectangles iff $n|a$ or $n|b$.* From the context, one must conclude that this statement is correct and should be proven."

[10]`github.com/openai/openai-cookbook/blob/main/techniques_to_improve_reliability.md`

## Acknowledgments

This work was partially supported by the AXA Research Fund. The authors would like to thank Karan Desai for helpful discussions and advise on the datasheet.

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

# Appendix

# A  Further Related Works

In this section, we present further related works. For (Chat)GPT, most investigations related to mathematical reasoning consist of anecdotal evidence concerning its performance and its failure modes. Notable mentions on social media can, for instance, be found in [5, 6, 41, 42]. Unfortunately, a clear methodology is missing, as most of the results are scattered on various internet platforms and cannot be easily reproduced.

To the best of our knowledge, the only investigations into the mathematical capabilities prior to the appearance of our first preprint were undertaken by [43, 44]. However, these works only report a small number of qualitative results, often on rather simple mathematical tasks and without specifying the precise versions of (Chat)GPT. The latter reference reports results only on a few selected examples, while the former reference investigates ChatGPT's[11] ability to compute irrational numbers as well as to solve some elementary math word problems.

Recently, the dataset by [45] contains a systematic evaluation of ChatGPT on the GSM8K dataset [26], the MATH dataset [12], and the MMMLU-STEM dataset [46]. These datasets allow for an automatic evaluation using only accuracy as an evaluation metric. The LĪLA dataset [47] also consists of a set of diverse mathematical problems, spanning different dimensions and problem types. The problems are formulated in such a way that solutions can be given by Python programs, which precludes problems that involve more advanced forms of mathematical proofs. Further anecdotal examples of mathematical performance are presented in [45].

Among LLMs prior to (Chat)GPT, Minerva [21], based on PaLM (discussed in Section 2), stands out, being trained in equal parts on websites that contain MathJax elements and arXiv preprints (additionally to general natural language data on which PaLM was trained). It achieves a score of roughly 50% on the significantly harder *Mathematics Aptitude Test of Heuristics* (MATH) dataset [12], which was sourced from various mathematical competitions. One distinguishing feature of the MATH dataset is that its problems admit a unique answer that can be condensed within a few characters (a number, for example). This is beneficial for the automatic evaluation of a model on such a dataset, since one can simply check the final answer, ignoring the step-by-step solution.

Finally, we would also like to mention the field of *formalized* mathematics, where large databases that encode advanced mathematical concepts exist, e.g., the *Lean Mathematical Library* [48]. Some of the ideas that we have used in this article, such as using prompts that formulate a task to fill in gaps in proofs, are echoed in [49] for datasets for formal mathematics consisting of expression trees. Yet, for the purpose of doing mathematics with large language models, these formal datasets cannot be leveraged since no straightforward way exists to convert them to natural language.

# B  Dataset Creation

## B.1  Categorization

Our subdatasets can be categorized along the following dimensions (see Table 1):

- **Mathematical difficulty (ascending):**

  $\boxed{M1}$ Elementary arithmetic problems, as found in the MATH dataset [12] at lower levels of difficulty.

  $\boxed{M2}$ Symbolic problems (integration of functions) that can also be solved via a supervised-learning, data-driven approach to mathematics [13].

  $\boxed{M3}$ (Under)graduate-level exercises from well-known textbooks [35–39] as well as questions from `math.stackexchange.com`, spanning diverse domains of mathematics.

  $\boxed{M4}$ Exercises that are in the style of mathematical olympiad problems, such as those taken from Engel's *Problem-Solving Strategies* book [40].

- **Question/prompt type:**

---

[11]Using an unknown version of ChatGPT that predates the 9-January-2023 version.

| $Q1$ | Review questions, which ask to state or name certain mathematical facts correctly. |

| $Q2$ | Overview-type review questions, which cut through an entire field of mathematics. |

| $Q3$ | Computational questions. |

| $Q4$ | Proof-based questions, which ask for a theorem proof or for a puzzle solution. |

| $Q5$ | Proof-completion questions, where a proof that has gaps or is incomplete needs to be completed. |

- **Types of high out-of-distribution likelihood:**

| $D1$ | Nontrivial problem encoding: The data points from the *Symbolic Integration* sub-dataset come from [13] and are publicly available[12]. Since the online training set uses Polish notation, it is very unlikely that (Chat)GPT has seen the corresponding prompts in LaTeX before. |

| $D2$ | Succinct solution: The solutions for the *Olympiad-Problem-Solving* subdataset are included in the book by Engel [40]. But the solutions are extremely concise, and simply repeating them would not show an immediate understanding of the problem. |

| $D3$ | Spoken dialogue: The *Search-Engine-Aspects* subdataset is unlikely to be well represented in the data on which (Chat)GPT has been trained since its prompts resemble word fragments that might appear in a mathematical dialogue (e.g., an oral mathematical exam), rather than in a textbook. |

## B.2 Format

The dataset consists of a collection of UTF-8 encoded JSON files. We explain the JSON keys of each data point in our dataset in the following and also indicate whether its value is optional. If the value is optional, the key has to be present, but the value will be an empty array or string.

- `prompt` denotes the input that we provide to (Chat)GPT through its web interface at the URL `chat.openai.com/chat`, see also Appendix C. We use a new session for each prompt to avoid (Chat)GPT being biased by previous prompts.

- `output` denotes the raw output that (Chat)GTP supplies us with. In some cases, mathematical formulas were rendered in the web interface such that we copied them in LaTeX.

- `rating` is a number from `1` to `5` that shows how many points (Chat)GPT has scored, `5` being a perfect answer and `1` being the lowest rating. A detailed explanation of the rating policy that we followed is contained in Appendix B.4.

- `errorcodes` *(optional)* highlight a list of error types that illustrate the failure modes of (Chat)GPT in a more fine-grained way. Not all types of errors apply to all (sub)datasets: For example, an error code for a missing proof step would not be applicable on a dataset that tests whether (Chat)GPT can multiply numbers or find prime divisors. The detailed explanation of the error codes (and the warning codes; see below) that was provided to the annotators is contained in Appendix B.4. There, we also include a policy of how ratings and error codes have to be used together.

- `warningcodes` *(optional)* highlight any problematic aspects of (Chat)GPT; for example, (Chat)GPT might be rambling and providing the user with unrelated information or use a poor (but correct) way of solving problems.

- `comment` *(optional)* denotes any noteworthy commentary that an assessor of (Chat)GPT may make. This can be used to give a more detailed explanation of the output, provide reasoning behind awarding a certain error code or rating, or generally provide context. For some subdatasets, this key was used to indicate the difficulty level of the prompt, as well

---

[12]`github.com/facebookresearch/SymbolicMathematics`

as an official solution, if available, see Section 3.1. It was also used to indicate whether we used prompt engineering, see Appendix 4.2.

- `msc` denotes a list of *mathematics subject classifications*[13] (MSC) that pertain to the output. Note that we do not classify the prompt given to (Chat)GPT with MSC codes, as there may be no proper classification; for example, when (Chat)GPT is asked what the most important theorem in all of mathematics is[14], it is meaningless to assign an MSC code to that prompt.

- `ref` *(optional)* indicates a reference to where the prompt was originally taken from (for some subdatasets, such as *Holes-in-Proofs*, we have used excerpts from various books or `math.stackexchange.com`; the original source was recorded as a value of this key). This key can have an empty value if the question was formulated by the authors and no authoritative source was plausible.

- `confidence` indicates how confident we have perceived (Chat)GPT to be when presenting us with its output. We allow values of `high`, `medium`, and `low`.

- `timestamp` denotes when the prompt was entered into (Chat)GPT. This can be used to track the version of (Chat)GPT; see Section 4.1.

The values of these keys within a single data point interact in nontrivial ways: If a rating of `5` is given, then it is expected that no error code is present—though there may be warning codes that are used. The error codes and warning codes are loosely in the spirit of a compiler throwing errors and warnings if it is given incorrect or sloppy code—although we have a role reversal, where the human is now the compiler, and the machine produces the code. In this sense, for some prompts, we have used multiple error and/or warning codes, which is why the corresponding values are arrays of strings. We use these codes to collect statistics on the behavior of (Chat)GPT; see Section 4.

For most of the subdatasets that make up our GHOSTS dataset, we have used LaTeX to encode mathematical formulas in our prompts. Our experiments have shown that (Chat)GPT can process LaTeX-encoded mathematics well.

The usage of MSC codes can be useful for mathematicians who want to integrate (Chat)GPT in their daily workflow, as it allows them to know in which areas the model performs better and can hence be trusted more. Our dataset is very diverse, having a total of 78 MSC codes. The top short versions of these codes (first two digits) are `26` ("Real functions", 127 occurrences) followed by `00` ("General", 110 occurrences) and `46` ("Functional analysis", 77 occurrences), see also Figure 5.

### B.3 Copyright and Licensing Terms

Because our dataset draws on multiple sources, which have different copyright holders, that have issued different licenses (or no licenses at all), we associate licenses on a fine-grained level to our data, to differentiate between parts where we are bound by the license of an existing dataset and the parts that we created, that we release under the following Creative Commons license: Attribution-NonCommercial 4.0 International (CC BY-NC 4.0)[15]. By this license, one may not use those parts of the dataset that were created by us for commercial purposes, and one must give appropriate credit when using it non-commercially; if users are building on the GHOSTS dataset, they need to indicate the changes that were made and distribute their contributions under the same license as the original.

**Licenses per datapoint:** We associate licenses on a fine-grained level to our dataset in the following way:

- For each datapoint, one license applies to the value of the `prompt` key (which, in some cases, is governed by copyright where no license is issued, in which case we do not release it publicly; in that case, the value of the prompt key will be `<copyrighted>`). Which `prompt` values are subject to which licenses is fully detailed in Table B.1.

---

[13]A complete list of MSC codes can be accessed at the URL `zbmath.org/static/msc2020.pdf`.
[14]The answer is Pythagoras' theorem, according to (Chat)GPT.
[15]See `https://creativecommons.org/licenses/by-nc/4.0/` for the detailed terms of the license.

Table B.1: A summary of all the files from all the subdatasets comprising our GHOSTS dataset, together with their license. In the case of the `prompt` values from the "Proofs Collection A" file, either no license was issued from the copyright holder (if the prompt value was taken from the books *Linear Algebra Done Right* by S. Axler or *Principles of Mathematical Analysis* by W. Rudin, cf. the `ref` key associated to that `prompt` value); or one of three Creative Commons Attribution-ShareAlike license, in different versions, depending on the date of the `math.stackexchange.com` post which served the source for the `prompt` value, see `https://math.stackexchange.com/help/licensing`.

| Subdataset Name | Comprised of the `json` file(s) | License for value of `prompt` key |
|---|---|---|
| ***G**rad-Text* | W. Rudin, Functional Analysis (ch. 1) | No license specified[17] |
| | W. Rudin, Functional Analysis (ch. 2) | No license specified[17] |
| | J. Munkres, Topology (ch. 1) | No license specified[17] |
| | J. Munkres, Topology (ch. 2) | No license specified[17] |
| | R. Durrett, Probability Theory | No license specified[18] |
| ***H**oles-in-Proofs* | Proofs Collection A | More than one license |
| | Proofs Collection B Prealgebra | MIT license |
| | Proofs Collection B Precalculus | MIT license |
| ***O**lympiad-Problem-Solving* | Olympiad Problem Solving | No license specified[17] |
| ***S**ymbolic-Integration* | Symbolic Integration | CC BY-NC 4.0 |
| *MA**T**H* | MATH Algebra | MIT license |
| | MATH Counting and Probability | MIT license |
| | MATH Prealgebra | MIT license |
| | MATH Precalculus | MIT license |
| ***S**earch-Engine-Aspects* | Definition Retrieval | CC BY-NC 4.0 |
| | Reverse Definition Retrieval | CC BY-NC 4.0 |
| | Named Theorem Proof Completion | CC BY-NC 4.0 |

- Another license, the CC BY-NC 4.0 license, applies to all other, remaining components of each datapoint; for the datapoints where we also created the prompt, these two licenses will coincide, as outlined by able B.1. We note that therefore, in particular, the value of the `output`, is also released by us under the CC BY-NC 4.0 license. This is consistent with OpenAI's terms, which at the time of writing state[16]: "*You may provide input to the Services ("Input"), and receive output generated and returned by the Services based on the Input ("Output"). Input and Output are collectively "Content." As between the parties and to the extent permitted by applicable law, you own all Input. Subject to your compliance with these Terms, OpenAI hereby assigns to you all its right, title and interest in and to Output. This means you can use Content for any purpose, including commercial purposes such as sale or publication if you comply with these Terms.*"

**Release policy:** Some of the subdataset files contain prompts that are protected under copyright, and no specific license has been issued for them by the copyright holder: These are all the prompts from all files from the *Grad-Text* and *Olympiad-Problem-Solving* subdatasets, as well as selected prompts from the "Proofs Collection A" file. In such cases where a copyrighted prompt was used for which no license was available, we have opted not to withhold the value of the `prompt` key in the publicly released dataset. The `ref` key includes a detailed reference to the page where the original theorem or exercise was presented, so a reader can easily retrieve the original prompt.

### B.4  Data Sources and Labeling Policies

Parts of the GHOSTS dataset were collected from books. These prompts were transcribed into LATEX. The output from (Chat)GPT's web interface was copied as-is, even if the output was not valid LATEX code. If the output contains rendered mathematical expressions, our

---

[16] `https://openai.com/policies/terms-of-use`

[17] Despite our best efforts to reach the copyright holder, we received no response, and, therefore, we do not have a license under which we can (re-)share the prompt.

[18] Permission was given by the author to reproduce the exercises we used, but unclarities remain who the copyright owner is.

policy was to transcribe it to LaTeX. Below are the policies that were followed by each assessor of (Chat)GPT's output regarding the rating, the error codes, and the warning codes:

**Rating**

- 1 → failure to understand the query (e.g., the user asks it something about number theory, and it responds with information about differential equations);

- 2 → query was understood, but the answer was entirely wrong (e.g., the user asks what the prime divisors of 111 are[19], and it responds with 8 and 6);

- 3 → query was understood, but the answer was only partially correct (e.g., the user asks it what the prime divisors of 111 are, and it responds with 3 and 6);

- 4 → query was understood, and the answer was mostly correct (e.g., the user asks it what the prime divisors of 222 are[20] and it responds with 3 and 37);

- 5 → query was understood and answer was completely correct.

**Error codes**

- e1 → missing examples or information (e.g., the user asks it what the prime divisors of 111 are, and it responds with 3, missing 37); this also applies, if (Chat)GPT ignores a part of the prompt (e.g., an equivalence needs to be shown, but (Chat)GPT shows only one direction);

- e2 → a few wrong/vague statements (e.g., the user asks it what the prime divisors of 30030 are[21] and it responds with 2, 3, 5, 7, 13 (wrong); or says that 2, 3, 5, and some other numbers are prime divisors (vague)); it can also denote a single statement, that is slightly vague;

- e3 → a lot of wrong/too vague statements (e.g., the user asks it what the prime divisors of 30030 are, and it responds with 2, 5, 8, 12, 13, 15 (wrong); or says that 2 and many other numbers are prime divisors (vague)); it can also denote a single statement, that is highly vague;

- e4 → wrong computations (i.e., an additional error flag to disambiguate between statements that are of computational nature or not);

- e5 → denotes wrong logic or wrong flow of arguments, which we further subdivide into specific flags, as we prohibit the use of e5 on its own (since it would be uninformative):

  - e5_1 → (Chat)GPT claims that to complete a proof, statements need to be shown that are unrelated to the claim;

  - e5_2 → a proof step is missing;

  - e5_3 → an edge case has not been considered;

  - e5_4 → an inference step is not supported (e.g., (Chat)GPT claims that from A follows B, but this claim is not true);

  - e5_5 → circular logical argument (i.e., using the hypothesis to prove the hypothesis);

- e6 → the general set-up is understood, but the legal operations are not respected or misunderstood (e.g., we are given a puzzle where we are only allowed to add even integers, but (Chat)GPT changes the rules and motivates the solution by allowing the addition of odd integers; or (Chat)GPT misunderstands an adjective that has multiple mathematical meanings, such as "dual", which can mean either topological dual space or algebraic dual space).

The following policy applies for error codes: If a rating $r$ with $1 < r < 5$ has been given, then an error code is mandatory to explain the type of error that occurred. For a perfect score of 5, no error codes should be assigned (but warning codes can be assigned). If the

---

[19]They are 37 and 3.
[20]They are 2, 37, and 3.
[21]They are 2, 3, 5, 7, and 11.

score is lowest, i.e., a rating of `1`, error codes can be assigned, but do not have to: In the case where (Chat)GPT has not understood the prompt, there typically is no reason to further detail the type of error.

**Warning codes**

- `w1` → (Chat)GPT is withholding essential information related to the prompt (e.g., the user asked it something about the integral $\int_{-\infty}^{\infty} e^{-x^2}\, dx$, and it answers correctly but does not tell the user that the integral was actually a famous, named integral, i.e., the Gaussian integral);

- `w2` → (Chat)GPT is rambling (i.e., after answering, correctly or incorrectly, (Chat)GPT tells the user much more details than the user wanted to know);

- `w3` → (Chat)GPT is hallucinating (i.e., after answering, correctly or incorrectly, (Chat)GPT tells the user unrelated information);

- `w4` → (Chat)GPT behaves weirdly (e.g., by using a weird proof structure (where applicable), using strange mathematical formulations, or by adopting a strange tone of the conversation or making opinionated statements);

- `w5` → (Chat)GPT changes the notation from the prompt without being instructed to do so (e.g., the prompt contains a vector space $\mathbb{X}$, but (Chat)GPT calls it $\mathbb{F}$).

### B.5  Mitigating Human Errors

Any assessment procedure that has a human component is prone to introducing bias—in particular, a procedure involving manual work such as rating the model outputs. In the list below we describe safeguards that help to mitigate bias and human errors (such as typos) as well as streamlining procedures that we undertook:

1. **Safeguards against LaTeX errors and typos:**

   Various typographical errors may appear due to incorrect LaTeX formatting. In this case, we noticed that (Chat)GPT was able to correctly infer what was intended (e.g., `$cup$` was correctly interpreted as `$\cup$`). Similarly, we noticed that (Chat)GPT's output was stable under small "perturbations" of the prompt, such as minor typos. Thus, (Chat)GPT therefore provided its own safeguard against these types of errors.

2. **Safeguards against encoding issues:**

   We presented clear instructions to each author who prompted (Chat)GPT on how to record and save the data in order to avoid any file encoding issues. In the end, all JSON files were inspected and streamlined to Unicode, if a different encoding was used.

3. **Safeguards against unfair comparisons:**

   Clear instructions were given to all authors that used (Chat)GPT to ensure that the language model has, to the extent possible, an identical state and starts from a blank chat.

4. **Safeguards against missing data and copy-paste errors:**

   Given a lack of API access in the early stages of our investigation (see Appendix C), there was a fair amount of data being copied from (Chat)GPT. To mitigate any copy-paste errors, several passes over the entire dataset, as well as automatic checks, were made to look, e.g., for potential inconsistencies, missing timestamps, and outputs not matching the prompts.

5. **Safeguards against dataset misformatting**:

   Because the methodology authors had to adhere to is complex (see Appendix B.2 and B.4, we introduced a number of automatic checks to make sure the general format of each datapoint is consistent and in line with our rules. In particular, we have made sure that various entries are in the correct place and take the correct values (e.g., that no warning flag values are in the key for errors, that the `ref` key value has the correct format, etc.).

6. **Safeguards against other unforeseen errors**:

   *Random samples*: Random samples ($< 10$) were drawn from each dataset, and a second assessor reviewed the rating. If deemed problematic, the original assessors were asked to re-evaluate.

   *Statistical checks*: Additional statistical checks were carried out as plausibility checks to make sure no other unforeseen errors occurred: If prompts deviated from the average length on that dataset, they were flagged, the output was manually inspected, and, if deemed necessary, a re-evaluation was carried out.

We are aware that these measures are not exhaustive, but given a fixed time budget, we considered them the most feasible. We note that authors evaluated ChatGPT models in multiple stages, as the models became available, and that checks and corrections were made at the end.

Because our checks and corrections pipelines are complex and a number were made after all the evaluations on all models were obtained, we cannot exclude that the strings containing the prompts weren't altered in very minor ways (e.g., converting back to Unicode from a different, original, unknown encoding may introduce slight variations of certain special characters that slipped through inspection). However, because of the sufficiently large number of evaluations that we carried out in each file, which is the smallest unit on which we collect statistics, this will not affect any results. From the viewpoint of reproducing our dataset, this is also not an issue, since even for identical prompts, (Chat)GPT itself is non-deterministic; this continues even if its `temperature` parameter in the API is set to zero [50, 51]. Lastly, as noted above, (Chat)GPT's answers will almost always still express the same idea, even if minor variations of the same prompt are used, so even individual scores will stay unchanged in such a scenario.

## B.6 Human Effort

The evaluation was carried out by a subset of the authors of this paper who have substantial mathematical expertise, ranging from master's degrees in mathematics to postdoc-level and professor-level positions at departments of mathematics. Assignment of prompts was done based on mathematical difficulty, with more senior mathematicians having received more difficult prompts. No third parties were involved.

Each of the 709 prompts of the GHOSTS dataset was evaluated on both the 9-January-2023 and 30-January-2023 version of ChatGPT; an additional 24 prompts were used to test the effect of prompt engineering on a single type of subdataset, see Appendix 4.2. We further evaluate GPT-4 on the 170 prompts of the miniGHOSTS dataset. This amounts to a total of 1636 prompt evaluations of advanced mathematics performed by graduate-level researchers.

We like to mention that our effort has occasionally unearthed small inconsistencies in existing datasets: For example, the "MATH Counting and Probability" file, which was sourced from the larger MATH dataset [12], contains the prompt "*What is the value of* $101^3 - 3 \cdot 101^2 + 3 \cdot 101 - 1$*?*", which is neither about counting, nor about probability, but arithmetic (our MSC codes allow users to find such examples).

We note that it is neither possible to outsource the creation of these subdatasets to a crowdsourcing service, such as Amazon Mechanical Turk, nor is it possible to generate them automatically from code because advanced mathematical insight is required for the creation of each prompt (where applicable) and for providing the fine-grained evaluation of the mathematical capabilities. We note that already in the case of the MATH dataset, which contained less advanced mathematics than some of our subdataset contained, it was noted in [12]: "Because MATH requires a strong mathematical background to perform well on, and a long amount of time to solve problems, we were restricted to assessing six human participants and could not rely on crowdsourcing sites such as Amazon Mechanical Turk". Furthermore, unlike in the case of the MATH dataset, the answer to most of our prompts cannot be condensed into a few tokens (such as a number or a function), e.g., when the answer is a mathematical proof.

This raises the difficulty of the creation of more data since graduate-level (and in some cases, PhD-level) mathematics is required. The combined effort of devising mathematically

insightful prompts and carefully rating the output of (Chat)GPT amounts to 1636 prompt evaluations, totaling several hundreds of person-hours, see Appendix B.6. However, as a result of these efforts, our dataset goes beyond all the mentioned mathematical datasets for LLMs in Section 2 in terms of the different aspects of mathematical reasoning that are being tested.

## B.7 Dataset Misuse

Our dataset does not contain personally identifiable information. Because the prompts of our dataset are compiled from a diverse collection, it does not contain systematic collections of mathematical exercises and their answers from a single type of test, which could invalidate certain school-level or university-level tests. We are not aware of any direct way of misusing our dataset, as its primary objective is the detailed evaluation of the mathematical reasoning performance of LLMs.

# C  Background details on (Chat)GPT

GPT-4, launched on 1st March 2023, is the latest model of the GPT lineage [7], being the successor of various versions of ChatGPT, the first of which was launched on 30 November 2022 [11]. These are all based on InstructGPT, which in turn is based on a trained GPT-3 [27], and fine-tuned using reinforcement learning with human feedback [52].

We note that already for models that predate (Chat)GPT, such as InstructGPT, where research articles and model cards [53] have been released, full reproducibility is not possible since the code and exact datasets have not been released. Furthermore, it was confirmed by OpenAI employees that for some of their models, launched prior to 30 November, a slight mismatch exists between the trained model that is accessible via the OpenAI web interface and the model referred to in the official paper [54]. This indicates how essential it is to document carefully which model our analysis pertains to and how we have accessed it. In our dataset, we have accordingly included time stamps for each prompt in order to be able to track, based on information provided by OpenAI, any changes in (Chat)GPT's version that have occurred.

We have exclusively used the GUI web interface to carry out the evaluation. This was necessary for consistency reasons since, at the beginning of our evaluation, API access was not yet widely available. At the time of writing, API access to GPT-4 is still limited, and a waitlist is employed, which made the use of the GUI web interface a necessity for GPT-4 [55]). We note that there exist no official documents that link the GUI web interface to the different model versions and possible model settings from the API. The 9-January-2023 and 30-January-2023 ChatGPT versions we evaluated are likely to be earlier instances of the newer model `gpt-3.5-turbo-0301`. This model itself is a "*snapshot of `gpt-3.5-turbo` from March 1st, 2023*" that will not receive further updates [56]. For GPT-4, at the time of writing, there exist four models `gpt-4`, `gpt-4-0314`, `gpt-4-32k`, `gpt-4-32k-0314` that can be used for the chat completion API endpoint [57]. Additionally, for all models, there exist various settings when using the chat completion API endpoint, such as `temperature` or `presence_penalty` that influence the models' output, which cannot be controlled via the GUI web interface. It is also not known which values of these settings are used for the GUI web version. The only version identifier in the GUI web version is a generic "model version" link at the bottom of the page that links to the release notes [11]. The 9-January-2023 and 30-January-2023 model versions that we evaluated are the ones presented in the release notes [11] at the respective time.

# D  Further Results

## D.1  Qualitative Analysis of Subdatasets on ChatGPT 9-January-2023

In this section, we go through common mistakes performed by ChatGPT, as well as notable observations regarding the output, one subdataset at a time. We focus on the 9-January-2023 version, see Section D.4 for more information regarding the other version. We note that the output of (Chat)GPT (and, generally, LLMs) is stochastic and therefore may differ on the same prompt. Nonetheless, clear trends can be observed, which we describe here. Individual outputs can be found in Appendix G.

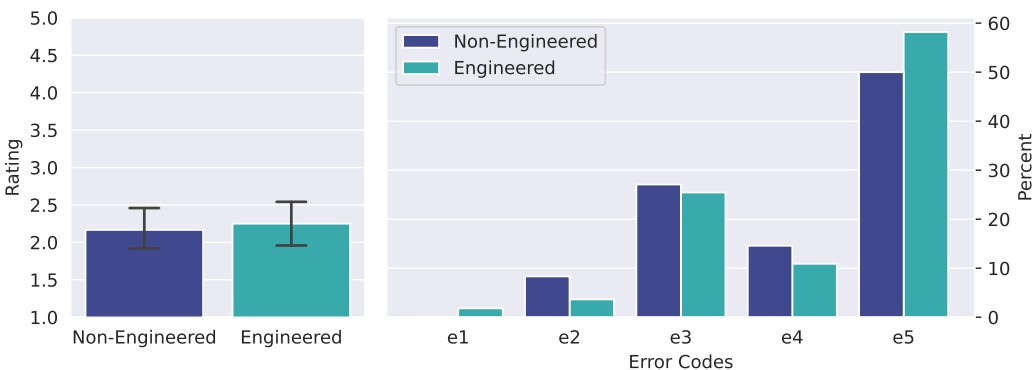

Figure 3: Effect of prompt engineering on the rating (left) and the error codes (right) for the 9-January-2023 model.

**Grad-Text**  ChatGPT, version 9-January-2023, performed best on simple set-theory and logic questions (the first chapter from the book *Topology* by J. Munkres [36]), which is reflected in its rating, see Figure 1. On the rest of the books, it performed substantially worse. Because of the confidence (`high`) with which it outputs the answer, the use of ChatGPT, version 9-January-2023, is particularly deceiving in this use-case since it may be intensively used by students studying these subjects.

**Holes-in-Proofs**  ChatGPT, version 9-January-2023, correctly recognized most well-known results or concepts (e.g., filling in the mean-value theorem, given a proof that lacked a reference to it). However, the ability of ChatGPT to execute algebraic manipulations is surprisingly inconsistent. In some cases, ChatGPT executes complicated symbolic tasks with ease; in other cases, it fails on simple arithmetic operations or rearranging terms. The mistakes do not seem to correlate with the complexity of the algebraic expression. When ChatGPT makes an algebraic mistake, it mostly carries over this mistake reliably to the rest of the computation.

**Olympiad-Problem-Solving**  On this subdataset, ChatGPT, version 9-January-2023, performed the poorest. From a mathematical point of view, these questions were also by far the most difficult, as they can pose difficulties even to professional mathematicians. A score of 3 was awarded when the answer started to show promise. However, 75% of the scores are 2 because the answer does not show any promise. No rating of 5 was awarded, and only one rating of 4 was achieved. This version of ChatGPT had a tendency to try and solve many questions using induction arguments. While this is not necessarily false, this was very far from the solutions given in the book, and this version's inductive proofs were easily seen to contain mistakes. In addition, ChatGPT often had difficulty understanding unconventional puzzles. For example, in the questions involving changing the color of squares on a chessboard, the solution offered by ChatGPT did not cover an $8 \times 8$ chessboard. Sometimes it tried to solve the problem by changing only 5 squares, far from the 32 required. Similarly, the 9-January-2023 version of ChatGPT struggled to respect unusual constraints in the questions, resulting in 8 `e6` errors, the highest number of `e6` errors out of all subdatasets. In some cases where the problem seemed to require complicated mathematics but was actually solvable by elementary techniques, ChatGPT did not spot this but instead referred to the general theory of, e.g., diophantine equations. Interestingly, ChatGPT would sometimes say, e.g., that the question could be solved with these means but that this was hard, so the confidence score was downgraded in these cases to `medium` or `low`.

**Symbolic-Integration**  The 9-January-2023 version of ChatGPT was dominated by systems that were trained specifically to solve integration problems [13]. In a number of instances, this version got the structure of terms right (for example, the number of summands in the output, as well as where factors had to be placed before summands), but it failed at concrete computations. Even very simple examples were not correct. For example, the antiderivative of $x \mapsto x^2/2$ is evaluated to $x \mapsto x^3/3 + C$, where $C$ is a constant of integration (the correct answer being $x \mapsto x^3/6 + C$). For a number of prompts, this version claims there

is no closed-form solution for the integral with complete confidence when, in fact, there is a solution; only integrals that have an elementary antiderivative are in this dataset.

**MATH** On the questions related to Algebra and Probability theory, the 9-January-2023 version of ChatGPT got the reasoning often correctly. However, the most common type of error was `e4`, occurring 36% of the time (in total 62 times). This version of ChatGPT may struggle when confronted with standard operations, such as inverting fractions, least common multiples, and changing the sign of numbers when moving them from one side of the equal sign to the other. Often, in these questions, a correct solution requires performing multiple operations in sequence. In such cases, most often, at least one operation was wrong. This prevented the model from getting a rating of `5` on the output, which was only achieved for 29% of the questions.

**Search-Engine-Aspects** On the *Search-Engine-Aspects* file, the 9-Januar-2023 version of ChatGPT knew almost all the theorems that it was asked at a basic level but made mistakes when stating them. When it came to listing other results required for the proofs, this version typically requested way more than the necessary theory—occasionally even results that only follow from the theorem which was asked for (error code `e5_5`). On the *Definition Retrieval* file, this version had quite a good performance: it recited most definitions correctly. It sometimes got confused when being asked about distributions in the sense of elements of the dual space of test functions. ChatGPT, version 9-January-2023, strongly favors the notion of distributions in the stochastic sense. Similarly, for the adjective "closed", where it chose to pick the context of algebra (instead of topology) and interpreted it to mean "algebraically closed". On the *Reverse Definition Retrieval* file, this version had the strongest performance, being able to recover most definitions from their descriptions, with an average rating of `4.30` (standard deviation `1.14`). This indicates the usefulness of ChatGPT as a general-purpose mathematical search engine. This subdataset is also the simplest from a mathematical point of view since no logical thinking is required, but only a name needs to be found.

## D.2 (Chat)GPT's Confidence

(Chat)GPT is usually very confident, unlike other LLMs that we have experimented with. As an illustrative example, consider the following prompt testing the sensitivity to LaTeX-encoded mathematics vs. Unicode-encoded mathematics:

```
Here are two statements: "Let (X, A, μ) be a measure space. Show that
the signed measures λ: A -> R form a Banach space."; "Let $(X,A,\mu)$
be a measure space. Show that the signed measures $\lambda:A
\rightarrow R$ form a Banach space." Are they equivalent?
```

The response by ChatGPT is not phrased in order to show any nuance in terms of confidence (which is typical, even if ChatGPT is wrong):

```
Yes, the two statements are equivalent. They both express the same
mathematical idea but use different notation and formatting
conventions. The first statement uses parentheses to enclose the
elements of the measure space and regular letters to represent the
functions, while the second statement uses subscripted parentheses
and uses the LaTeX formatting to represent the elements of the
measure space and the function.
```

The response by Codex [58], another model that we briefly tested (but whose scope would have exceeded that of a single conference article) gives a cautions response and, unlike ChatGPT, is capable of voicing doubt:

```
I think they are equivalent, but I am not sure.
```

## D.3 Figures of ChatGPT's Performance (version 9-January-2023)

In this section, we collect figures that extend the discussion in the main body and provide further views on the data and descriptive statistics.

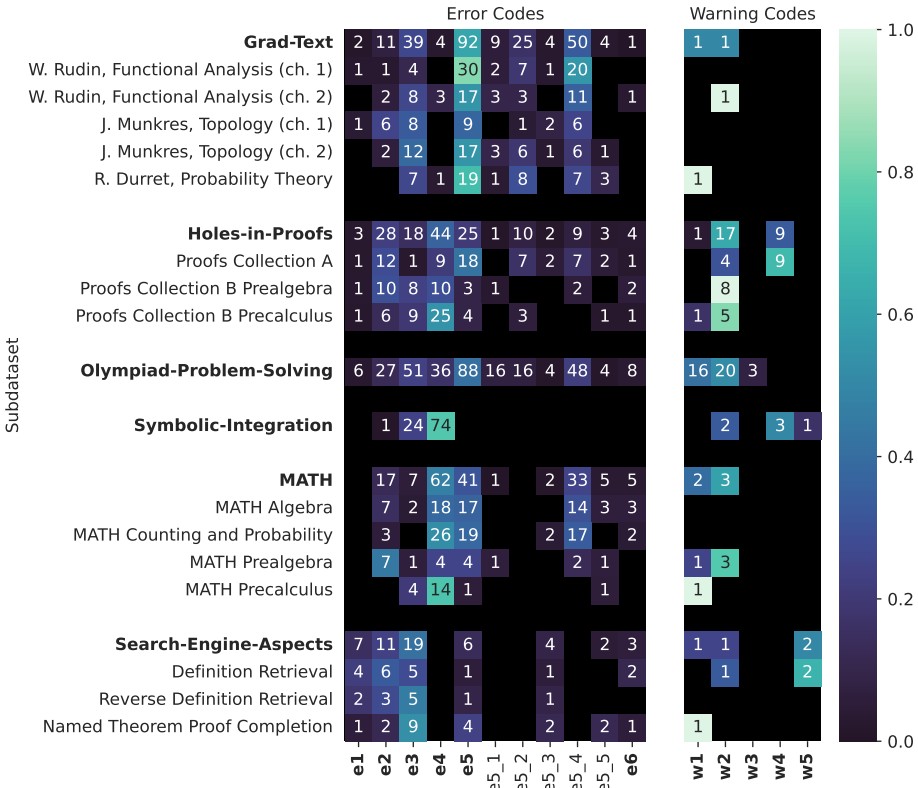

Figure 4: Counts (annotation) and relative frequencies (color) of error and warning codes by subdatasets (bold) and files for ChatGPT 9-January-2023 on GHOSTS.

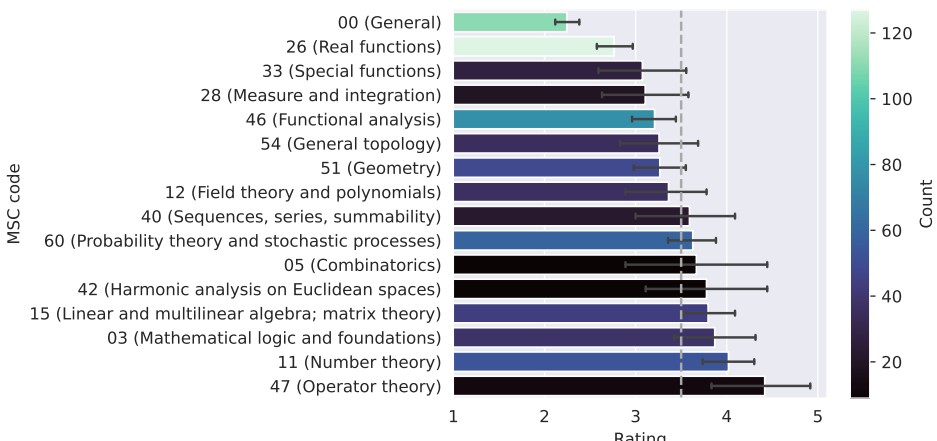

Figure 5: Average rating over mathematical fields for the 9-January-2023 version of ChatGPT on GHOSTS. The color depicts the occurrence of each MSC code, and only MSC codes that have at least 5 occurrences are shown. Note that the ranking is not indicative of the complexity of the fields since we do not use equally complicated exercises for all fields. The error bars represent 95% confidence intervals.

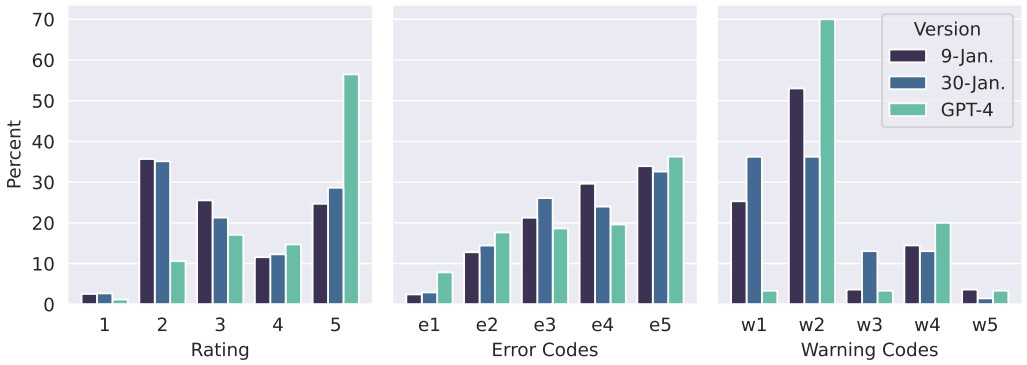

Figure 6: A comparison of the 9-January-2023 model, the 30-January-2023 model (both on GHOSTS), and GPT-4 (on miniGHOSTS) in terms of percentages of ratings (right), error codes (middle), and warning codes (right).

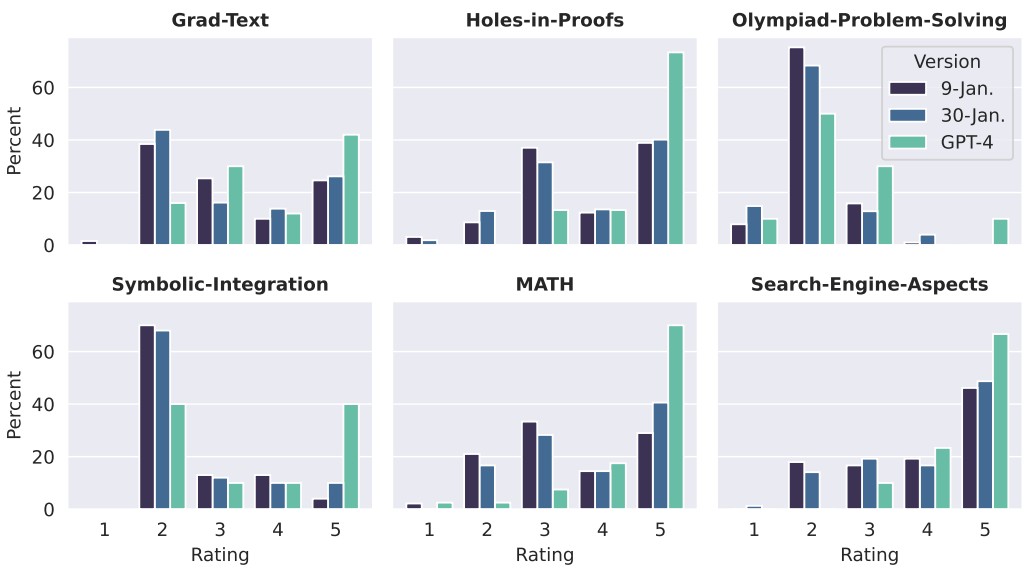

Figure 7: A comparison of the 9-January-2023 model, the 30-January-2023 model (both on GHOSTS), and GPT-4 (on miniGHOSTS) in terms of percentages of ratings on the different subdatasets.

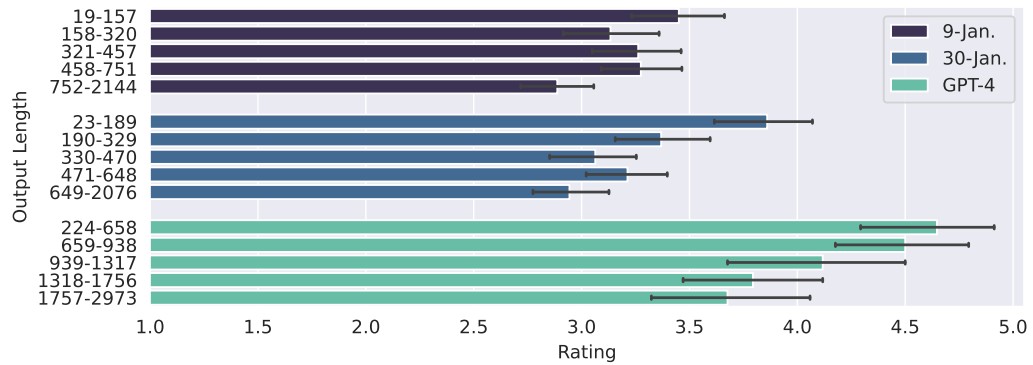

Figure 8: A comparison of the 9-January-2023 model, the 30-January-2023 model (both on GHOSTS), and GPT-4 (on miniGHOSTS) in terms of output lengths. Every interval contains 20% of the prompts, and the error bars represent 95% confidence intervals.

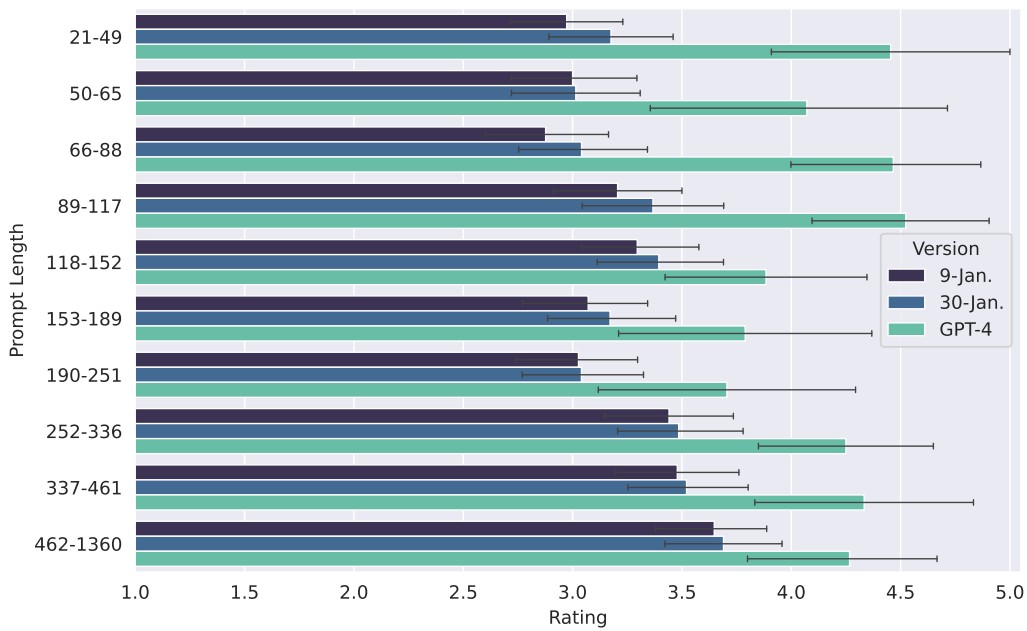

Figure 9: A comparison of the 9-January-2023 model, the 30-January-2023 model (both on GHOSTS), and GPT-4 (on miniGHOSTS) in terms of prompt lengths. Every interval contains 10% of the prompts of GHOSTS, and the error bars represent 95% confidence intervals.

## D.4 Comparison of (Chat)GPT Versions

In this section, we collect figures which illustrate the differences and similarities between versions of (Chat)GPT. We note that even though the 30-January-2023 version performs very similarly to the 9-January-2023 version, there are some differences in the distribution of ratings, error codes, and warning codes, see Figure 6.

On the other hand, GPT-4 strictly dominates the ChatGPT versions in terms of performance. It always provides context around the question (whether that was asked for or not) and often gives useful (and correct) pointers that, for example, highlight the importance of a particular theorem. Figure 8 depicts the verbosity of different (Chat)GPT versions and the achieved rating. However, we also note that the optimal level of verbosity can depend on the mathematical background of the user. As a result, there have been significantly more warning codes of type `w2` (i.e., rambling) for GPT-4, see Figure 6.

## E   The miniGHOSTS and microGHOSTS Dataset

The miniGHOSTS and microGHOSTS datasets consist of a subset of datapoints from the GHOSTS dataset. Their role is to enable faster evaluation of a given language model on advanced mathematics by shrinking the dataset size from 709 prompts (GHOSTS) to 170 prompts (miniGHOSTS), which represent about 24% of GHOSTS and 14 prompts (microGHOSTS), which represent about 2% of GHOSTS.

The miniGHOSTS dataset was created by using a heuristic to select a subset of prompts such that the mean and variance of the 30-January-2024 version of ChatGPT on this subset match the mean and variance of the 30-January-2024 version of ChatGPT on the entire GHOSTS dataset. The role of the miniGHOSTS dataset is to allow a full assessment of a model on a reduced dataset. This dataset is, therefore, not meant to assess the mathematical reasoning performance of other LLMs in absolute terms but rather to provide a relative comparison to the 30-January-2023 version of ChatGPT.

The prompts of the microGHOSTS were chosen by hand from the worst-performing questions from miniGHOSTS, or those that were deemed to pose mathematical difficulties to language models. The role of microGHOSTS is to obtain a preliminary assessment of a model by reducing the human evaluation cost to close to zero (to have initial information before potentially deciding on embarking on a more comprehensive evaluation).

Our overall evaluation experience leads us to observe that there exist the following broad categories that lead to ChatGPT performing problematically:

- instances where complex mathematical definitions are involved, which depend in turn on prior definitions (e.g., an absorbing subset of a topological vector space), and where something is asked about how these concepts interact;

- instances where the reasoning is predominantly geometric reasoning or where an insight is obtained by thinking geometrically;

- long chains of computations;

- vague definitions that require ChatGPT to understand that terms in mathematics are overloaded;

- problems when constructing examples;

- lack of ingenious ideas, such as needed for Olympiad-problem solving.

We have leveraged these insights to include prompts that specifically exploit these weaknesses.

Prompts that belong to the microGHOSTS dataset we tag by using the `<microghosts>` word as a value in the `comment` key. Further, the values of the `comment` also contain extensive solutions and explanations of the problem so that a good gold standard exists against which a given LLM solution can be compared by a human rater. We also include general explanations of the mathematical concepts that are involved to further aid general machine learning practitioners (who may not have prior mathematical training) if they want to test their language models' reasoning capabilities.

# F    Limitations and Reproducibility

In this section, we describe the limitations of what our dataset and our evaluations cover; some of these also are related to issues regarding reproducibility, which we also discuss below.

In terms of:

- **model selection**, we have focused on those models that, at the time of writing, are known to have the best performance among existing LLMs, either based on anecdotal evidence [3, 5, 6] or via previous benchmarks on more elementary mathematical reasoning tasks [7]. Because human evaluation has been very time-consuming, we have chosen to invest the available time solely in the evaluation of these models (and not older ones as well) to obtain a picture of their performance that is as accurate as possible.

- **correctness**, we have put a large number of measures in place to guarantee evaluations are as correct as possible (see Appendix B.5). Nonetheless, we cannot guarantee the absolute correctness of each evaluation. There are also some questions where, aside from human errors, fundamental evaluator disagreement occurs regarding the question of what rating should be given; this is a well-known problem in psychometrics and, in a simple instance, can be resolved by having multiple evaluators evaluate the same question. Given a fixed budget in terms of man-hours, we have chosen not to follow this approach because 1) we opted to be able to devise a larger dataset and 2) in light of the checks that we described in Appendix B.5, such an approach would result in diminishing returns on invested time.

An issue in this regard is the fact that an absolute ground truth is lacking for a mathematical dataset of this type that goes beyond simple arithmetical questions. In particular, in the case of proofs, these can be presented in myriad ways: On one hand, for the same mathematical fact, one can find conceptually distinct proofs. (See [59] for an extreme example of this, where 122 conceptually distinct proofs of the Pythagorean theorem are given.) On the other hand, one and the same proof can be presented in different ways, since the order in which the arguments are made and connected can be changed[22].

Assuming all the previous problems were, somehow, solvable, further complicating the matters is the fact that not all possible proofs are known, and new ones are being discovered for known statements (staying with the topic of the Pythagorean theorem, even millennia after its publication, new proofs have surfaced [60]). If a model were to output a proof that is not known, it would still require a human to judge correctness, as in this case, there may not be any ground truth available to compare against.

All of the above makes it very hard for the ground-truth data to be included in our dataset - and if it is included, it may not be meaningful for the outlined reasons. Some elementary datasets, such as the MATH dataset [12], have ground-truth data included in the comment. However, their ground truth is also susceptible to the issues we mentioned; human evaluation is necessary for absolute certainty.

(Chat)GPT's non-determinism, ground-truth absence, and author disagreements notwithstanding, the above shows that an effort to reproduce our results should lead to highly similar scores on each file of each subdataset. Due to the fact that our analysis is made per subdataset file and not per prompt, we are confident that a double-digit number of evaluations per file makes the conclusions that we draw robust.

- **prompt engineering type**, we have deliberately chosen to avoid more complex types of prompt engineering mechanics, such as using Chain-of-Thoughts [61], or Tree-of-Thoughts [62] because 1) the main focus of the paper is the standard performance of the tested language models, not their the performance augmented by various forms of in-context learning and 2) these types of prompt engineering methods require additional significant human effort to decorate the original prompts with example when carrying out few-shot prompting and to track the output. The human effort for the current dataset was already considerable; adding such prompt engineering methods would have further

---

[22]E.g., if proving statements $C$ depends on three statements $X,Y,Z$, we could prove first $X,Y,Z$ and then conclude $C$; but we could also prove $Y,Z,X$ and then conclude $C$ - or construct the proof based on any other permutation of $X,Y,Z$. While all of these proof presentations would use the same ideas, syntactically, they would be different proof presentations

raised the evaluation cost while also increasing the complexity of our non-trivial evaluation methodology and 3) for GPT-3 and certain datasets and prompting engineering techniques, performance was degraded [63].

- **datapoint-level MSC code coverage**, our dataset covers 78 distinct codes spanning most areas of mathematics. Because there are 1636 distinct, MSC-classified outputs, by the pigeonhole principle, some of the MSC codes will necessarily be covered only by a small amount of MSC code (Figure 5 indicates those MSC codes that occur at least five times, although some MSC codes appear over one hundred times).

  We also note that for particularly easy mathematical questions (e.g., simple arithmetical questions), no suitable MSC codes exist to classify the output since MSC codes typically classify more advanced mathematics[23]. Nonetheless, we have attempted to match them as well as possible and allow multiple MSC codes in order to classify the output as precisely as possible.

  We further note that an exhaustive survey of (Chat)GPT's performance across *every* MSC code would necessitate a large, community-driven effort to set up an extensive database. Due to the high cost of rating each output, requiring specialized skills, this is something that no individual research group could reasonably do – but we hope that our approach is a starting point for such an effort.

- **subdataset-level attribute tag coverage**, we could, in theory, start to investigate every combination of these attribute tags for the subdatasets. We recall that the subdatasets have attribute tags assigned to them that indicate difficulty to subdatasets, the type of questions a subdataset contains, and the ways the subdataset is likely to not have been part of training distributions of the considered models (see the explanation of the tags $M1$-$M4$, $Q1$-$Q5$ and $D1$-$D3$ from Appendix B.1 and Table 1). Thus, for a single mathematical domain, represented by one MSC code (such as for elementary arithmetic problems), one could generate subdatasets to cover every possible combination of the 60 possible ways to combine the attribution tag listed above. If one combines this with different MSC codes, this would, as above, also lead to a manual curation effort, which is too intensive and time-consuming for a single research group.

---

[23]The MSC codes starting with the numbers "97", which at first glance might be most suitable, are solely reserved to classify content that is related to the educational process of mathematics, rather than the mathematical content itself.

## G  Best-3 and Worst-3 Across Selected Subdatasets

We list below the best and worst answers of ChatGPT, version 9-January-2023, over a selection of subdatasets. For readability, the prompts and answers are lightly modified so that the LaTeX-based formulas are correctly displayed, and whitespace is removed.

### G.1  Grad-Text

Examples from the *Grad-Text* subdataset, comprising the books *Probability Theory* by R. Durret [35], *Topology* by J. R. Munkres [36] and *Functional Analysis* by W. Rudin [37].

**Best-3**

---

`"prompt"`: "Let $\mu$ be a measure, $\|f\|_\infty = \inf\{M : \mu(\{x : |f(x)| > M\}) = 0\}$ and $\|g\|_1 = \int |g| d\mu$. Prove that $\int |fg| d\mu \leq \|g\|_1 \|f\|_\infty$.",

`"output"`: "Let $M = \|f\|_\infty$ and $\epsilon > 0$. Then, by the definition of $\|f\|_\infty$, we have $\mu(\{x : |f(x)| > M + \epsilon\}) = 0$. Therefore,

$$\int |fg| d\mu = \int_{|f| \leq M+\epsilon} |fg| d\mu + \int_{|f| > M+\epsilon} |fg| d\mu \leq (M+\epsilon) \int |g| d\mu + \int_{|f| > M+\epsilon} |f||g| d\mu.$$

The second term on the right side is zero because $\mu(x : |f(x)| > M + \epsilon) = 0$. So we have $\int |fg| d\mu \leq (M+\epsilon) \int |g| d\mu = (M+\epsilon) \|g\|_1$. As $\epsilon$ can be arbitrarily small, we can let it go to zero and we get $\int |fg| d\mu \leq M \|g\|_1 = \|f\|_\infty \|g\|_1$. Hence proved."

---

`"prompt"`: "If $\{T_\alpha\}$ is a family of topologies on $X$, show that $\bigcap_\alpha T_\alpha$ is a topology on $X$.",

`"output"`: "To show that $\bigcap_\alpha T_\alpha$ is a topology on $X$, we need to show that it satisfies the three axioms of a topology: $\emptyset$ and $X$ are both in $\bigcap_\alpha T_\alpha$ since $\emptyset$ and $X$ are in every topology $T_\alpha$. If $\{U_i\}_{i \in I}$ is a collection of sets in $\bigcap_\alpha T_\alpha$, then $\bigcup_{i \in I} U_i$ is also in $\bigcap_\alpha T_\alpha$. Since each $T_\alpha$ is a topology, $\bigcup_{i \in I} U_i$ is in every $T_\alpha$, and so it is also in $\bigcap_\alpha T_\alpha$. If $\{U_i\}_{i \in I}$ is a collection of sets in $\bigcap_\alpha T_\alpha$, then $\bigcap_{i \in I} U_i$ is also in $\bigcap_\alpha T_\alpha$. Since each $T_\alpha$ is a topology, $\bigcap_{i \in I} U_i$ is in every $T_\alpha$, and so it is also in $\bigcap_\alpha T_\alpha$. Therefore, $\bigcap_\alpha T_\alpha$ is a topology on $X$."

---

`"prompt"`: "Let $X$ be a vector space and $A$ and $B$ subsets of $X$. If $A$ and $B$ are balanced, so is $A + B$.",

`"output"`: "A subset $S$ of a vector space $X$ is called balanced if for every vector $x$ in $X$ and every scalar $\lambda$ with $|\lambda| \leq 1$, $\lambda x$ is in $S$ if $x$ is in $S$. To prove that $A + B$ is balanced, we must show that for every vector $x = a + b$ in $A + B$, and every scalar $\lambda$ with $|\lambda| \leq 1$, $\lambda x = \lambda(a + b) = \lambda a + \lambda b$ is in $A + B$ if $x$ is in $A + B$. Since $A$ and $B$ are balanced, we know that for every $a$ in $A$ and every scalar $\lambda$ with $|\lambda| \leq 1$, $\lambda a$ is in $A$ if $a$ is in $A$. Similarly, for every $b$ in $B$ and every scalar $\lambda$ with $|\lambda| \leq 1$, $\lambda b$ is in $B$ if $b$ is in $B$. So, if $x = a + b$ is in $A + B$, then $a$ is in $A$ and $b$ is in $B$. Therefore, $\lambda a$ is in $A$ and $\lambda b$ is in $B$, and thus $\lambda x = \lambda a + \lambda b$ is in $A + B$. Therefore, $A + B$ is balanced."

---

**Worst-3**

---

`"prompt"`: "Find a function $f : \mathbb{R} \to \mathbb{R}$ that is continuous at precisely one point.",

`"output"`: "One example of such a function is

$$f(x) = \begin{cases} x^2 & \text{if } x \neq 0 \\ 1 & \text{if } x = 0. \end{cases}$$

This function is continuous at $x = 0$ but not at any other point."

---

"prompt": "Let $X$ be a topological vector space. All sets mentioned
    below are understood to be the subsets of $X$. Prove the following
    statement: If $A$ is compact and $B$ is closed, then $A+B$ is
    closed.",
"output": "A topological vector space is a vector space with a
    topology such that the vector addition and scalar multiplication
    are continuous. To prove that $A+B$ is closed, we must show that
    the complement of $A+B$ is open. Let $y \in X \setminus (A+B)$. Then there
    exists some $x_1 \in A$ and $x_2 \in B$ such that $y = x_1 + x_2$. Since $A$ is compact
    and $B$ is closed, there exist neighborhoods $U_1$ of $x_1$ and $U_2$ of $x_2$
    such that $U_1$ is relatively compact and $X \setminus U_2$ is open. Now consider
    the set $V = U_1 + U_2 = \{x_1 + x_2 : x_1 \in U_1, x_2 \in U_2\}$. This set is an open
    neighborhood of $y$ and it is contained in $X \setminus (A+B)$, which shows
    that $X \setminus (A+B)$ is open and hence $A+B$ is closed."

---

"prompt": "Let $X$ be a topological vector space. All sets mentioned
    below are understood to be the subsets of $X$. Prove the following
    statement: The sum of two closed sets may fail to be closed.",
"output": "A counterexample can be used to prove that the sum of two
    closed sets may fail to be closed. For example, let $X = \mathbb{R}$ and let
    $A = [0,1]$ and $B = (1,2]$. Both $A$ and $B$ are closed sets in $\mathbb{R}$, but
    $A + B = [0,2]$ is not a closed set in $\mathbb{R}$."

---

## G.2   Holes-in-Proofs (Proofs Collection A)

Examples from the *Holes-in-Proofs* subdataset, *Proofs Collection A* file, based on the
books [38, 39] and questions from math.stackexchange.com

**Best-3**

"prompt": "We want to show that any open subset of $\mathbb{R}$ is a countable
    union of disjoint open intervals. Let $U$ be a non—empty open
    subset of $\mathbb{R}$. For $x, y \in U$ define $x \sim y$ iff [MISSING]. It's easily
    checked that $\sim$ is an equivalence relation on $U$ whose equivalence
    classes are pairwise disjoint open intervals in $\mathbb{R}$. (The term
    \enquote{interval} here includes unbounded intervals, i.e.,
    rays.) Let $\mathscr{I}$ be the set of $\sim$—classes. Clearly $U = \bigcup_{I \in \mathscr{I}} I$. For
    each $I \in \mathscr{I}$ choose a rational $q_I \in I$; the map $\mathscr{I} \to \mathbb{Q} : I \mapsto q_I$ is
    injective, so $\mathscr{I}$ is countable. Can you fill in the missing
    condition?",
"output": "The missing condition is "there exists an open interval
    containing $x$ and $y$ that is contained in $U$." "

---

"prompt": "The $p$—norm on $\mathbb{R}^n$ is given by $\|x\|_p = \left( \sum_{k=1}^n |x_k|^p \right)^{1/p}$. We want to
    show show that $\|x\|_p \le C \|x\|_q$ for arbitrary $1 \le p < q$. We compute

$$\|x\|_p = \left( \sum_{i=1}^n |x_i|^p \right)^{1/p} \le \left( \left( \sum_{i=1}^n |x_i|^q \right)^{\frac{p}{q}} n^{1 - \frac{p}{q}} \right)^{1/p} = \left( \sum_{i=1}^n |x_i|^q \right)^{\frac{1}{q}} n^{\frac{1}{p} - \frac{1}{q}} = n^{1/p - 1/q} \|x\|_q.$$

Which result has been used in the inequality above?",
"output": "The inequality used in the above calculation is Hölder's
    inequality. It states that for any $p, q \ge 1$ such that $\frac{1}{p} + \frac{1}{q} = 1$, and
    for any vectors $x \in \mathbb{R}^n$, $y \in \mathbb{R}^n$,

$$\left( \sum_{i=1}^n |x_i y_i| \right) \le \left( \sum_{i=1}^n |x_i|^p \right)^{1/p} \left( \sum_{i=1}^n |y_i|^q \right)^{1/q}."$$

"prompt": "Let $f$ be a real−valued function continuous on $[a, b]$ and differentiable on $(a, b)$. Suppose that $\lim_{x \to a} f'(x)$ exists. Then, we want to prove that $f$ is differentiable at $a$ and $f'(a) = \lim_{x \to a} f'(x)$. Using the definition of the derivative, we need to show that

$$\lim_{h \to 0^+} \frac{f(a+h) - f(a)}{h}$$

exists and is equal to $\lim_{x \to a^+} f'(x)$. Note that for $h > 0$ the Mean Value Theorem provides a point $c_h$ with $a < c_h < a + h$ such that [MISSING]. Finally, note that $c_h \to a^+$ as $h \to 0^+$. Can you fill in the missing equation?",

"output": "The missing equation from the Mean Value Theorem is:

$$f'(c_h) = \frac{f(a+h) - f(a)}{h}.\text{"}$$

## Worst-3

"prompt": "We want to compute

$$\sum_{n=1}^{\infty} \frac{2n}{3^{n+1}}.$$

Let's first find a general formula for the following sum: $S_m = \sum_{n=1}^{m} nr^n$. Notice that

$$S_m - rS_m = -mr^{m+1} + \sum_{n=1}^{m} r^n = -mr^{m+1} + \frac{r - r^{m+1}}{1 - r} = \frac{mr^{m+2} - (m+1)r^{m+1} + r}{1 - r}.$$

Hence

$$S_m = \frac{mr^{m+2} - (m+1)r^{m+1} + r}{(1-r)^2}.$$

Can you finish the calculation?",

"output": "Yes, to finish the calculation we can substitute the values of $m = \infty$, $r = 2/3$ into the formula we found for $S_m$:

$$S_\infty = \frac{\infty \left(\frac{2}{3}\right)^{\infty+2} - (\infty + 1) \left(\frac{2}{3}\right)^{\infty+1} + \frac{2}{3}}{\left(1 - \frac{2}{3}\right)^2}$$

This simplifies to

$$S_\infty = \frac{\frac{2}{9}}{\frac{1}{9}} = 2, \text{"}$$

"prompt": "Suppose $V$ is an inner product space and $T \in \mathcal{L}(V)$. We want to prove that

$$|\det(T)| = \det \sqrt{T^*T}.$$

By the Polar Decomposition there is an isometry $S \in \mathcal{L}(V)$ such that

$$T = S\sqrt{T^*T}.$$

Can you finish the proof?",

"output": "Yes, once we have that $T = S\sqrt{T^*T}$, we can take the determinant of both sides to get

$$\det(T) = \det(S) \det(\sqrt{T^*T}).$$

Since $S$ is an isometry, it preserves inner products, so $\det(S) = \pm 1$. Therefore,

$$|\det(T)| = |\det(S)||\det(\sqrt{T^*T})| = |\det(\sqrt{T^*T})|.$$

Since the determinant of a matrix is always non−negative, we have that $|\det(T)| = \det(\sqrt{T^*T})$"

"prompt": "$F$ is a finite field of order $q$. We want to compute the size of $\mathrm{GL}_n(F)$. In order for an $n \times n$ matrix to be invertible, we need the rows to be linearly independent. As you note, we have $q^n - 1$ choices for the first row; now, there are $q$ vectors in the span of the first row, so we have $q^n - q$ choices for the second row. Now, let $v_1, v_2$ be the first two rows. Then the set of vectors in the span of $v_1, v_2$ is of the form $\{c_1 v_1 + c_2 v_2 | c_1, c_2 \in F\}$. This set is of size $q^2$, as we have $q$ choices for $c_1$ and $q$ choices for $c_2$. Thus, we have $q^n - q^2$ choices for the third row. Continuing this gives the desired formula. Can you state the formula?",

"output": "The size of $\mathrm{GL}_n(F)$ is given by the formula

$$(q^n - 1)(q^{n-1} - q)(q^{n-2} - q^2) \cdots (q^2 - q)(q - 1).$$"

### G.3 Holes-in-Proofs (Proofs Collection B Prealgebra and Precalculus)

Examples from the *Holes-in-Proofs* subdataset, *Proofs Collection B Prealgebra* and *Proofs Collection B Precalculus* files, based on [12].

**Best-3**

"prompt": "Find the smallest positive integer that is greater than 1 and relatively prime to the product of the first 20 positive integers. Reminder: two numbers are relatively prime if their greatest common divisor is 1. Solution: <MISSING> the smallest possible integer is $\boxed{23}$. Fill in the missing steps.",

"output": "The first 20 positive integers are 1, 2, 3, 4, 5, 6, 7, 8, 9, 10, 11, 12, 13, 14, 15, 16, 17, 18, 19, and 20. The product of these numbers is 2432902008176640000. To find the smallest positive integer greater than 1 that is relatively prime to this product, we can start at 2 and incrementally check each number until we find one that has a greatest common divisor (GCD) of 1 with the product. We can check 2, 3, 4, ..., 22, and see that the GCD of each of these numbers with the product is not 1. However, when we check 23, we find that the GCD of 23 and 2432902008176640000 is 1. Therefore, 23 is the smallest positive integer that is greater than 1 and relatively prime to the product of the first 20 positive integers.",

"prompt": "Find the number of solutions to $\cos \frac{x}{4} = \cos x$ in the interval $0 < x < 24\pi$. Solution: From the equation $\cos \frac{x}{4} = \cos x$, $\cos x - \cos \frac{x}{4} = 0$. From the sum—to—product formula, we can write this as

$$-2 \sin \frac{5x}{8} \sin \frac{3x}{8} = 0.$$

<MISSING> Fill in the missing steps.",

"output": "Since the product of sines is equal to zero, either $\sin \frac{5x}{8} = 0$ or $\sin \frac{3x}{8} = 0$. Solving for each case: $\sin \frac{5x}{8} = 0 \implies \frac{5x}{8} = n\pi$, where $n$ is an integer. This gives us the solutions $x = \frac{8n}{5}\pi$. $\sin \frac{3x}{8} = 0 \implies \frac{3x}{8} = n\pi$, where $n$ is an integer. This gives us the solutions $x = \frac{8n}{3}\pi$. Now we need to find the solutions that are in the interval $0 < x < 24\pi$. $x = \frac{8n}{5}\pi$ gives solutions $x = \frac{16}{5}\pi, \frac{24}{5}\pi, \frac{32}{5}\pi, \frac{40}{5}\pi, \frac{48}{5}\pi$ which are in the given interval. $x = \frac{8n}{3}\pi$ gives solutions $x = \frac{8}{3}\pi, \frac{16}{3}\pi, \frac{24}{3}\pi$ which are in the given interval. Therefore, there are 5 solutions for $x = \frac{8n}{5}\pi$ and 3 solutions for $x = \frac{8n}{3}\pi$ in the given interval. The total number of solutions is 5+3=8."

"prompt": "Let $\mathbf{a}$, $\mathbf{b}$, $\mathbf{c}$ be three vectors such that $\|\mathbf{a}\| = \frac{1}{\sqrt{2}}$, $\|\mathbf{b}\| = \frac{1}{\sqrt{3}}$, $\|\mathbf{c}\| = \frac{1}{\sqrt{6}}$, $\|\mathbf{a} + \mathbf{b} + \mathbf{c}\| = 1$, and

$$\mathbf{a} = k(\mathbf{b} \times \mathbf{c})$$

for some scalar $k$. Find the angle between $\mathbf{b}$ and $\mathbf{c}$, in degrees.
Solution: Since $\mathbf{b} \times \mathbf{c}$ is orthogonal to both $\mathbf{b}$ and $\mathbf{c}$, $\mathbf{a} \cdot \mathbf{b} = 0$ and
$\mathbf{a} \cdot \mathbf{c} = 0$. Since $\|\mathbf{a} + \mathbf{b} + \mathbf{c}\| = 1$,

$$(\mathbf{a} + \mathbf{b} + \mathbf{c}) \cdot (\mathbf{a} + \mathbf{b} + \mathbf{c}) = 1.$$

<MISSING> Fill in the missing steps.",
"output": "Expanding the dot product on the left side gives:
$\mathbf{a} \cdot \mathbf{a} + \mathbf{a} \cdot \mathbf{b} + \mathbf{a} \cdot \mathbf{c} + \mathbf{b} \cdot \mathbf{a} + \mathbf{b} \cdot \mathbf{b} + \mathbf{b} \cdot \mathbf{c} + \mathbf{c} \cdot \mathbf{a} + \mathbf{c} \cdot \mathbf{b} + \mathbf{c} \cdot \mathbf{c} = 1$. Since
$\mathbf{a} \cdot \mathbf{b} = \mathbf{a} \cdot \mathbf{c} = \mathbf{b} \cdot \mathbf{a} = \mathbf{c} \cdot \mathbf{a} = 0$, this simplifies to: $|\mathbf{a}|^2 + |\mathbf{b}|^2 + |\mathbf{c}|^2 + 2\mathbf{b} \cdot \mathbf{c} = 1$. We
know that $|\mathbf{a}|^2 = \frac{1}{2}$, $|\mathbf{b}|^2 = \frac{1}{3}$, $|\mathbf{c}|^2 = \frac{1}{6}$, and substituting these values in
gives: $\frac{1}{2} + \frac{1}{3} + \frac{1}{6} + 2\mathbf{b} \cdot \mathbf{c} = 1$ Solving for $\mathbf{b} \cdot \mathbf{c}$ gives: $\mathbf{b} \cdot \mathbf{c} = \frac{1}{6} - \frac{1}{6} = 0$ Which
means the vectors are orthogonal, hence the angle between them is
90 degrees."

## Worst-3

"prompt": "The line described by

$$\begin{pmatrix} 2 \\ -1 \\ 3 \end{pmatrix} + t \begin{pmatrix} k \\ 2 \\ 1 \end{pmatrix}$$

is perpendicular to the line described by

$$\begin{pmatrix} 2 \\ -1 \\ 1 \end{pmatrix} + u \begin{pmatrix} 2 \\ 1 \\ 2 \end{pmatrix}$$

and passes through the point $(4, a, b)$. Find $a + b + k$. Solution: The
direction vector of the first line is $\begin{pmatrix} k \\ 2 \\ 1 \end{pmatrix}$, and the direction
vector of the second line is $\begin{pmatrix} 2 \\ 1 \\ 2 \end{pmatrix}$. <MISSING> Then $t = -1$, so $a = -3$
and $b = 2$, so $a + b + k = \boxed{-3}$. Fill in the missing steps.",
"output": "The direction vector of the first line is $\begin{pmatrix} k \\ 2 \\ 1 \end{pmatrix}$, and the
direction vector of the second line is $\begin{pmatrix} 2 \\ 1 \\ 2 \end{pmatrix}$. To find the point of
intersection, we can set the two parametric equations equal to
each other:

$$\begin{pmatrix} 2 \\ -1 \\ 3 \end{pmatrix} + t \begin{pmatrix} k \\ 2 \\ 1 \end{pmatrix} = \begin{pmatrix} 2 \\ -1 \\ 1 \end{pmatrix} + u \begin{pmatrix} 2 \\ 1 \\ 2 \end{pmatrix}$$

Solving for t,u we get t=−1,u=−3. So the point of intersection is
$\begin{pmatrix} 2 \\ -1 \\ 3 \end{pmatrix} - \begin{pmatrix} k \\ 2 \\ 1 \end{pmatrix} = (4, a, b)$. Given that, we get a = −3 and b = 2, then
$a + b + k = \boxed{-3}$.",

"prompt": "How many positive integers less than 100 and divisible
by 3 are also divisible by 4? Solution: If we start counting
from 1, every third consecutive integer is divisible by 3 and
every fourth consecutive integer is divisible by 4. <MISSING>
so there are $\boxed{8}$ multiples of 12 between 1 and 100. Fill in
the missing steps.",

"output": "The missing steps are: We need to find the multiples of 12, which are divisible by both 3 and 4. To find the first multiple of 12 less than 100, we divide 100 by 12 and round down to the nearest integer, which is 8. To find the last multiple of 12 less than 100, we divide 12 by 100 and round up to the nearest integer, which is 8. The number of multiples of 12 between 1 and 100 is the difference between the last and first multiple, which is $8 - 8 = 0$. So there are no positive integers less than 100 that are divisible by both 3 and 4.",

---

"prompt": "The complex numbers $5 + 10i$, $7 + 2i$, and $11 + 3i$ are three vertices of a rectangle. Find the fourth vertex. Solution: Let $a = 5 + 10i$, $b = 7 + 2i$, and $c = 11 + 3i$. Let $d$ be the fourth vertex. Note that
$$c - b = 4 + i$$
and
$$a - b = -2 + 8i = 2i(4 + i) = 2i(c - b).$$
Thus, the angle between the segment joining $a$ and $b$ and the segment joining $b$ and $c$ is $90°$. <MISSING> Fill in the missing steps.",

"output": "Since the angle between the segment joining $a$ and $b$ and the segment joining $b$ and $c$ is $90°$, it follows that the segment joining $c$ and $d$ is perpendicular to the segment joining $a$ and $b$. Since $c - b = 4 + i$, we have $c - d = -(b - d) = -(-2 + 8i) = 2 - 8i$. Therefore, $d = c - (2 - 8i) = 11 + 3i - (2 - 8i) = 9 - 5i$. So the fourth vertex is $(9, -5i)$."

## G.4 Olympiad-Problem-Solving

Examples from the *Olympiad-Problem-Solving* subdataset based on the book [40].

**Best-3**

"prompt": "$n$ numbers are written on a blackboard. In one step, you may erase any two of the numbers, say $a$ and $b$, and write, instead $(a + b)/4$. Repeating this step $n - 1$ times, there is one number left. Prove that, initially, if there were n ones on the board, at the end, a number which is not less than $1/n$ will remain.",

"output": "This can be proven by induction. Base case: $n = 2$. If there are two ones on the board, after one step, the remaining number is $(1 + 1)/4 = 1/2$, which is not less than $1/2$. Inductive case: Assume that the statement is true for $n = k$. Now consider $n = k + 1$. If there are $k + 1$ ones on the board, we can erase any two of them and write their average, which is not less than $1/k + 1$ by the induction hypothesis. Therefore, we can conclude that the statement holds for all $n$."

---

"prompt": "The integers $1, \ldots, n$ are arranged in any order. In one step you may switch any two neighboring integers. Prove that you can never reach the initial order after an odd number of steps.",

"output": "This problem can be proven using the concept of parity. Specifically, the parity of a permutation is the number of inversions (i.e. pairs of integers $(i, j)$ such that $i < j$ but the ith integer appears after the $j$th integer in the permutation) modulo $2$. It can be shown that the parity of a permutation remains unchanged after any single swap of neighboring integers. Since the initial ordering has zero inversions (i.e. it has even parity), it follows that an odd number of swaps will always yield a permutation with odd parity, and thus it cannot be the same as the initial ordering."

```
"prompt": "Let's answer this question step by step. Derive
    divisibility criteria for 9 and 11.",
"output": "Divisibility Criteria for 9: If the sum of digits of a
    number is divisible by 9, then the number is also divisible by 9.
    If the difference between the sum of digits in the odd positions
    and even positions is divisible by 9, then the number is also
    divisible by 9. Divisibility Criteria for 11: If the difference
    between the sum of digits in the odd positions and even positions
    is divisible by 11, then the number is also divisible by 11. If
    the alternating sum of digits (first minus second plus third
    minus fourth, etc.) is divisible by 11, then the number is also
    divisible by 11."
```

**Worst-3**

```
"prompt": "Assume an 8 × 8 chessboard with the usual coloring. You may
    repaint all squares (a) of a row or column (b) of a 2 × 2 square.
    The goal is to attain just one black square. Can you reach the
    goal?",
"output": "Yes, it is possible to attain just one black square on an
    8 × 8 chessboard. One way to do this is to repaint all the squares
    of the top—left 2 × 2 square, which will leave one black square in
    the bottom—right corner of the chessboard. Another way to do this
    is to repaint all the squares of the first column, which will
    leave one black square in the bottom—right corner of the
    chessboard.
```

```
"prompt": "Start with the positive integers 1,..., 4n − 1. In one move
    you may replace any two integers by their difference. Prove that
    an even integer will be left after 4n − 2 steps.",
"output": "We can prove this by mathematical induction. Base case:
    When n = 1, we have the integers 1, 2, 3, 4. We can perform the
    following moves: (1, 2) → 1, 3, 4; (3, 4) → 1, 3; (1, 3) → 2, an even integer.
    Inductive case: Assume that the statement is true for n = k. We
    will show that it is also true for n = k + 1. We have the integers
    1, 2, 3, ..., 4(k + 1) − 1. We can perform the following moves:
    (1, 2) → 1, 3, 4, ..., 4(k + 1) − 1; (3, 4) → 1, 3, ..., 4(k + 1) − 3; ...;
    (4k − 1, 4k) → 1, 3, ..., 4(k + 1) − 5. Now we have 4k − 2 integers left,
    1, 3, ..., 4(k + 1) − 5, which are 4k − 2 consecutive odd integers. By the
    induction hypothesis, after 4k − 2 steps, we will be left with an
    even integer. Therefore, by mathematical induction, we can prove
    that an even integer will be left after 4n − 2 steps."
```

```
"prompt": "Let's answer this question step by step. Assume an 8 × 8
    chessboard with the usual coloring. You may repaint all squares
    (a) of a row or column (b) of a 2 × 2 square. The goal is to attain
    just one black square. Can you reach the goal?",
"output": "Yes, it is possible to reach the goal of having just one
    black square on an 8 × 8 chessboard. One way to do this is to
    repaint all of the squares in the first row and first column
    black, and then repaint the top—left square of the 2 × 2 square
    white. This results in one black square and all other squares
    white."
```

## G.5   Symbolic-Integration

Examples from our *Symbolic-Integration* subdataset based on [13].

**Best-3**

---

`"prompt"`: "What is the integral of $4x^7 + \sin(1+x)$",
`"output"`: "The integral of $4x^7$ is $(4/8)x^8 + C$, where $C$ is the constant of integration. The integral of $\sin(1+x)$ is $-\cos(1+x) + C$. So the integral of $4x^7 + \sin(1+x)$ is $(4/8)x^8 - \cos(1+x) + C$."

---

`"prompt"`: "What is the integral of $2048 + 32x^3$",
`"output"`: "The antiderivative or indefinite integral of $2048 + 32x^3$ with respect to $x$ is $\int(2048 + 32x^3)dx = 2048x + (32/4)x^4 + C$ where $C$ is the constant of integration."

---

`"prompt"`: "What is the integral of $12 + 6\cosh(x)$",
`"output"`: "The integral of $12 + 6\cosh(x)$ is $12x + 6\sinh(x) + C$, where $C$ is the constant of integration."

---

**Worst-3**

---

`"prompt"`: "What is the integral of $(4+x)[4+3x][e^{x*(4+x)^2}]$",
`"output"`: "The integral of $(4+x)[4+3x][e^{x*(4+x)^2}]$ is not expressible in terms of elementary functions."

---

`"prompt"`: "What is the integral of $3x - 4x^2[\cos(x+3)]$",
`"output"`: "The antiderivative of $3x - 4x^2 is x^2 - (4/3)x^3$ and the antiderivative of $[\cos(x+3)]is -\sin(x+3)$. Therefore, the integral of $3x - 4x^2[\cos(x+3)]$ is $x^2 - (4/3)x^3 - \sin(x+3) + C$, where $C$ is the constant of integration."

---

`"prompt"`: "What is the integral of $1/(2 + \ln(3/2))$",
`"output"`: "The integral of $1/(2 + \ln(3/2))$ cannot be expressed in terms of elementary functions and must be approximated numerically or solved using special functions."

---

# H  Datasheet for the GHOSTS Dataset

This appendix provides a datasheet for the GHOSTS dataset. The format of this datasheet was introduced in [64] and consolidates the motivation, creation process, composition, and intended uses of our dataset as a series of questions and answers.

## H.1  Motivation

Q1. **For what purpose was the dataset created?** *Was there a specific task in mind? Was there a specific gap that needed to be filled? Please provide a description.*

The existing datasets of natural-language mathematics are far from covering all the typical tasks professional mathematicians encounter in daily life, making it unclear whether language models can be of any help in this regard. Existing datasets mostly cover elementary mathematics or resemble standard tests like SATs (see Sections 2 and 3). Hence, they do not offer any insight into the usage of ChatGPT as a tool for mathematicians. In this work, we have made the first attempt towards filling this gap, going beyond math problems that are yes-no rated, and proposed a benchmark made and curated by working researchers in the field that tests different dimensions of mathematical reasoning.

Q2. **Who created this dataset (e.g., which team, research group) and on behalf of which entity (e.g., company, institution, organization)?**

The authors of this work created GHOSTS; see Appendix B.6 for more information.

Q3. **Who funded the creation of the dataset?** *If there is an associated grant, please provide the name of the grantor and the grant name and number.*

There is no associated grant or funding which has been used to create the GHOSTS dataset.

Q4. **Any other comments?**

No.

## H.2  Composition

Q5. **What do the instances that comprise the dataset represent (e.g., documents, photos, people, countries)?** *Are there multiple types of instances (e.g., movies, users, and ratings; people and interactions between them; nodes and edges)? Please provide a description.*

GHOSTS consists of textual prompts, in natural language, representing mathematical questions. For each prompt, GHOSTS contains one or more instances of outputs of (Chat)GPT and corresponding fine-grained evaluation by the authors.

Q6. **How many instances are there in total (of each type, if appropriate)?**

There are 709 prompts in GHOSTS; a selection of 170 of these makes up miniGHOSTS, and a selection of 14 of those makes up microGHOSTS. For 24 of the questions from the GHOSTS dataset, light prompt engineering variations have been carried out. Each of the $709 + 24$ questions from GHOSTS has been evaluated on ChatGPT, version 9-January-2023 and 30-January-2023, and 170 questions from miniGHOSTS have been evaluated on GPT-4 (and thus on microGHOSTS too). Thus, in total, $(709 + 24) \times 2 + 170 = 1636$ outputs and evaluations have been carried out. See also Appendix B.6 for more information.

Q7. **Does the dataset contain all possible instances or is it a sample (not necessarily random) of instances from a larger set?** *If the dataset is a sample, then what is the larger set? Is the sample representative of the larger set (e.g., geographic coverage)? If so, please describe how this representativeness was validated/verified. If it is not representative of the larger set, please describe why not (e.g., to cover a more diverse range of instances because instances were withheld or unavailable).*

GHOSTS tries to cover a wide range of mathematical questions from 78 different MSC codes; see Appendix B.1 and B.2. However, due to the prohibitive cost of human evaluation, which cannot be fully automated away (see Section 3.3), it is not feasible to

represent all mathematical fields across all dimensions of "mathematical behavior" and all types of mathematical questions (overview questions, fact-stating questions, etc.).

Q8. **What data does each instance consist of? "Raw" data (e.g., unprocessed text or images) or features?** *In either case, please provide a description.*

GHOSTS, miniGHOSTS, and microGHOSTS consist of a collection of JSON objects (one for each data point), and each JSON object consists of 10 key-values pairs as detailed in Appendix B.2.

Q9. **Is there a label or target associated with each instance?** *If so, please provide a description.*

No, we do not explicitly define a label or target for the instances. However, the `rating` of the output can potentially be used to select good and bad mathematical conversations of (Chat)GPT in order to fine-tune models and the `errorcodes` and `warningcodes` can be used to make a more fine-grained classification possible.

Q10. **Is any information missing from individual instances?** *If so, please provide a description, explaining why this information is missing (e.g., because it was unavailable). This does not include intentionally removed information but might include, e.g., redacted text.*

No.

Q11. **Are relationships between individual instances made explicit (e.g., users' movie ratings, social network links)?** *If so, please describe how these relationships are made explicit.*

Relations between instances are explicitly given by the same values on (subsets) of the fields, e.g., the same prompt, the same model version, or the same MSC code. Prompt-engineered variations of the same question are represented as an array of JSON objects, one object for each variation.

Q12. **Are there recommended data splits (e.g., training, development/validation, testing)?** *If so, please provide a description of these splits, explaining the rationale behind them.*

Not applicable.

Q13. **Are there any errors, sources of noise, or redundancies in the dataset?** *If so, please provide a description.*

The evaluation of the prompts included in GHOSTS underlies human errors. However, we tried to mitigate these errors; see Appendix B.5.

Q14. **Is the dataset self-contained, or does it link to or otherwise rely on external resources (e.g., websites, tweets, other datasets)?** *If it links to or relies on external resources,*
*(a) Are there guarantees that they will exist, and remain constant, over time?*
*(b) Are there official archival versions of the complete dataset (i.e., including the external resources as they existed at the time the dataset was created)?*
*(c) Are there any restrictions (e.g., licenses, fees) associated with any of the external resources that might apply to a future user? Please provide descriptions of all external resources and any restrictions associated with them, as well as links or other access points, as appropriate.*

The dataset is self-contained. However, some of the prompts cannot be publicly released since they are taken or adapted from sources that are protected by copyright, for which no license was given, see Appendix B.3; though we do release the output of the models on these prompts, together with our evaluation of the output. Further, we provide a reference to the original, copyrighted materials, so that a user can easily retrieve the original prompts.

Q15. **Does the dataset contain data that might be considered confidential (e.g., data that is protected by legal privilege or by doctor-patient confidentiality, data that includes the content of individuals non-public communications)?** *If so, please provide a description.*

No.

Q16. **Does the dataset contain data that, if viewed directly, might be offensive, insulting, threatening, or might otherwise cause anxiety?** *If so, please describe why.*

No.

Q17. **Does the dataset relate to people?** *If not, you may skip remaining questions in this section.*

No.

Q18. **Does the dataset identify any subpopulations (e.g., by age, gender)?** *If so, please describe how these subpopulations are identified and provide a description of their respective distributions within the dataset.*

No.

Q19. **Is it possible to identify one or more natural persons, either directly or indirectly (i.e., in combination with other data) from the dataset?** *If so, please describe how.*

No.

Q20. **Does the dataset contain data that might be considered sensitive in any way (e.g., data that reveals racial or ethnic origins, sexual orientations, religious beliefs, political opinions or union memberships, or locations; financial or health data; biometric or genetic data; forms of government identification, such as social security numbers; criminal history)?** *If so, please provide a description.*

No.

Q21. **Any other comments?**

No.

## H.3 Collection Process

Q22. **How was the data associated with each instance acquired?** *Was the data directly observable (e.g., raw text, movie ratings), reported by subjects (e.g., survey responses), or indirectly inferred/derived from other data (e.g., part-of-speech tags, model-based guesses for age or language)? If data was reported by subjects or indirectly inferred/derived from other data, was the data validated/verified? If so, please describe how.*

We collected and constructed prompts from various sources, see Table 1 and Section 3. For the evaluation, we captured the corresponding outputs of (Chat)GPT and rated them according to the instructions in Appendix B.2 and B.4.

Q23. **What mechanisms or procedures were used to collect the data (e.g., hardware apparatus or sensor, manual human curation, software program, software API)?** *How were these mechanisms or procedures validated?*

To query (Chat)GPT, we used the GUI web interface at the URL `chat.openai.com/chat`; see Appendix B.6 for detailed reasons for using the GUI interface.

Q24. **If the dataset is a sample from a larger set, what was the sampling strategy?**

The prompts of the MATH and Symbolic-Integration subdatasets have been randomly sampled from [12] and [13], across different files from those datasets.

The prompts taken from all the other sources (books and `math.stackexchange.com`, similarly have been randomly selected.

For our miniGHOSTS dataset, we sampled 10 prompts from each of the 17 files in GHOSTS in the following way: Our results in Section 4 indicate that the 9-January-2023 and the 30-January-2023 ChatGPT versions have similar overall performance; however, the behavior differs on a more fine-grained level and was marginally better for the 30-January-2023 version. Hence, we assembled miniGHOSTS by computing all subsets of 10 prompts having approximately the same mean rating and standard deviation as the original file from GHOSTS, rated on the 30-January-2023 version

of ChatGPT. A manual inspection of these subsets, in order to pick a subset with appropriate mathematical content (we want to have a mathematically diverse dataset), then led to the final selection of the miniGHOSTS dataset. The microGHOSTS dataset was created from manual inspection of the miniGHOSTS dataset, by isolating in total 14 questions which were deemed difficult for language models.

Q25. **Who was involved in data collection process (e.g., students, crowd-workers, contractors) and how were they compensated (e.g., how much were crowd-workers paid)?**

Only we have been involved in the data collection process. No payment (other than one made through regular employment) in relation to creating this dataset and writing this article was made.

Q26. **Over what timeframe was the data collected? Does this timeframe match the creation timeframe of the data associated with the instances (e.g., recent crawl of old news articles)?** *If not, please provide a description of the timeframe.*

The collection date matches the creation time. It is specified in the `timestamp` key in each data point from GHOSTS and spans a timeframe from January 9, 2023, to now. Using the timestamp, the version of ChatGPT that was used can be inferred, see Appendix C.

Q27. **Were any ethical review processes conducted (e.g., by an institutional review board)?** *If so, please provide a description of these review processes, including the outcomes, as well as a link or other access point to any supporting documentation.*

Not applicable.

Q28. **Does the dataset relate to people?** *If not, you may skip remaining questions in this section.*

No.

Q29. **Did you collect the data from the individuals in question directly, or obtain it via third parties or other sources (e.g., websites)?**

Not applicable.

Q30. **Were the individuals in question notified about the data collection?** *If so, please describe (or show with screenshots or other information) how notice was provided, and provide a link or other access point to, or otherwise reproduce, the exact language of the notification itself.*

Not applicable.

Q31. **Did the individuals in question consent to the collection and use of their data?** *If so, please describe (or show with screenshots or other information) how consent was requested and provided, and provide a link or other access point to, or otherwise reproduce, the exact language to which the individuals consented.*

Not applicable.

Q32. **If consent was obtained, were the consenting individuals provided with a mechanism to revoke their consent in the future or for certain uses?** *If so, please provide a description, as well as a link or other access point to the mechanism (if appropriate).*

Not applicable.

Q33. **Has an analysis of the potential impact of the dataset and its use on data subjects (e.g., a data protection impact analysis) been conducted?** *If so, please provide a description of this analysis, including the outcomes, as well as a link or other access point to any supporting documentation.*

Not applicable.

Q34. **Any other comments?**

No.

### H.4 Preprocessing, Cleaning, and/or Labeling

Q35. **Was any preprocessing/cleaning/labeling of the data done (e.g., discretization or bucketing, tokenization, part-of-speech tagging, SIFT feature extraction, removal of instances, processing of missing values)?** *If so, please provide a description. If not, you may skip the remainder of the questions in this section.*

We corrected various minor issues and inconsistencies that could arise in the process of manual evaluation, see Appendix B.5.

Q36. **Was the "raw" data saved in addition to the preprocessed/cleaned/labeled data (e.g., to support unanticipated future uses)?** *If so, please provide a link or other access point to the "raw" data.*

The `output` key in each JSON object contains the raw output from (Chat)GPT—unless ChatGPT used rendered LaTeX in which case our policy was to transcribe it. In very few cases, potential copy-paste errors were noticed, which were fixed.

Q37. **Is the software used to preprocess/clean/label the instances available?** *If so, please provide a link or other access point.*

The raw output of (Chat)GPT in the `output` key has not been cleaned, see Q36. Cleaning of the other values has been done first using Python scripts, in an automated way, and subsequently by hand, to correct any further, unforeseen mistakes, see Appendix B.5. The Python scripts are available upon request.

Q38. **Any other comments?**

No.

### H.5 Uses

Q39. **Has the dataset been used for any tasks already?** *If so, please provide a description.*

We have used the GHOSTS dataset to evaluate and compare the mathematical capabilities of different LLMs, in particular, different (Chat)GPT versions; see Section 4.

Q40. **Is there a repository that links to any or all papers or systems that use the dataset?** *If so, please provide a link or other access point.*

Future work citing the GHOSTS dataset will be listed by citation trackers such as Google Scholar and Semantic Scholar.

Q41. **What (other) tasks could the dataset be used for?**

If the dataset is growing further, we anticipate that GHOSTS can be used as training data for fine-tuning LLMs.

Q42. **Is there anything about the composition of the dataset or the way it was collected and preprocessed/cleaned/labeled that might impact future uses?** *For example, is there anything that a future user might need to know to avoid uses that could result in unfair treatment of individuals or groups (e.g., stereotyping, quality of service issues) or other undesirable harms (e.g., financial harms, legal risks) If so, please provide a description. Is there anything a future user could do to mitigate these undesirable harms?*

No.

Q43. **Are there any tasks for which the dataset should not be used?** *If so, please provide a description.*

No.

Q44. **Any other comments?**

No.

### H.6 Distribution

**Q45. Will the dataset be distributed to third parties outside of the entity (e.g., company, institution, organization) on behalf of which the dataset was created?** *If so, please provide a description.*

Yes, the GHOSTS, miniGHOSTS, and microGHOSTS datasets will be made publicly available. It consists of three JSON files, one per ChatGPT version (9-January-2023, 30-January-2023, and GPT-4, where the latter was evaluated on miniGHOSTS, while the former two on the full GHOSTS dataset; the microGHOSTS prompts are tagged within the miniGHOSTS dataset, see the `comment` key). Some prompts will not be available due to copyright issues (see Appendix B.3), but a precise reference where the original prompt can be found will be included instead.

**Q46. How will the dataset be distributed (e.g., tarball on website, API, GitHub)** *Does the dataset have a digital object identifier (DOI)?*

The dataset will be made available on GitHub in the public repository `github.com/friederrr/GHOSTS` as a collection of JSON files.

**Q47. When will the dataset be distributed?**

The dataset is already available.

**Q48. Will the dataset be distributed under a copyright or other intellectual property (IP) license, and/or under applicable terms of use (ToU)?** *If so, please describe this license and/or ToU, and provide a link or other access point to, or otherwise reproduce, any relevant licensing terms or ToU, as well as any fees associated with these restrictions.*

We release the GHOSTS, miniGHOSTS, and microGHOSTS datasets under the following Creative Commons license: Attribution-NonCommercial 4.0 International (CC BY-NC 4.0), unless we are bound by licenses of individual prompts or files from various subdatasets to release those prompts or files under more restrictive licenses; see Appendix B.3 for more information.

**Q49. Have any third parties imposed IP-based or other restrictions on the data associated with the instances?** *If so, please describe these restrictions, and provide a link or other access point to, or otherwise reproduce, any relevant licensing terms, as well as any fees associated with these restrictions.*

IP-restrictions apply only to those prompts that were not solely created by the authors (which are under the CC BY-NC 4.0, as explained above), see Appendix B.3 for these cases.

**Q50. Do any export controls or other regulatory restrictions apply to the dataset or to individual instances?** *If so, please describe these restrictions, and provide a link or other access point to, or otherwise reproduce, any supporting documentation.*

No.

**Q51. Any other comments?**

No.

### H.7 Maintenance

**Q52. Who will be supporting/hosting/maintaining the dataset?**

The dataset will be hosted on a GitHub repository; see Q46. All the information about the dataset, including links to the paper and future announcements, will be written in the README file of the GitHub repository.

**Q53. How can the owner/curator/manager of the dataset be contacted (e.g., email address)?**

The email addresses of the authors are publicly available. Moreover, it is possible to raise an issue on GitHub.

Q54. **Is there an erratum?** *If so, please provide a link or other access point.*

Future changes will be documented in the README file of the GitHub repository. Differences in single files can be tracked in the Git history.

Q55. **Will the dataset be updated (e.g., to correct labeling errors, add new instances, delete instances)?** *If so, please describe how often, by whom, and how updates will be communicated to users (e.g., mailing list, GitHub)?*

We will continue to maintain the dataset to fix any errors that may have occurred. We will allow either direct email contact or GitHub pull requests in this sense. Beyond fixing such errors, we consider the dataset to be frozen.

Q56. **If the dataset relates to people, are there applicable limits on the retention of the data associated with the instances (e.g., were individuals in question told that their data would be retained for a fixed period of time and then deleted)?** *If so, please describe these limits and explain how they will be enforced.*

Not applicable.

Q57. **Will older versions of the dataset continue to be supported/hosted/maintained?** *If so, please describe how. If not, please describe how its obsolescence will be communicated to users.*

Not applicable.

Q58. **If others want to extend/augment/build on/contribute to the dataset, is there a mechanism for them to do so?** *If so, please provide a description. Will these contributions be verified? If so, please describe how. If not, why not? Is there a process for communicating/distributing these contributions to other users? If so, please provide a description.*

Any external contribution to our dataset is strongly encouraged. Every addition to the dataset will be carefully reviewed by the authors. For other details, please see Q55.

