# OpenReview forum: "Mathematical Capabilities of ChatGPT"
_NeurIPS.cc/2023/Track/Datasets_and_Benchmarks — NeurIPS 2023 Datasets and Benchmarks Poster_

### Official Review · Reviewer_3gvx · 2023-07-10
**Review of "Mathematical Capabilities of ChatGPT"**

**Rating:** 5
**Confidence:** 4
**Clarity:** Yes

**Strengths:**

1） The paper provides a thorough evaluation of ChatGPT and GPT-4, considering their performance in different mathematical tasks.

2） The exploration of mathematical capabilities in language models is a timely and relevant topic. Since many papers and studies are focusing on the mathematical reasoning ability of LLMs, which is a critical ability for humans.


**Additional Feedback:**

NA

**Correctness:**

The shared datasets have no ‘ground-truth’ for problems, only the human evaluation on the GPT outputs.  This makes it difficult for future  usage. For the same problem, GPT may give different answers if running several times. Another group of annotators  may have different rating results.

**Documentation:**

The dataset was manually constructed. Reproducibility is unknown.  For example,  the rating key, from 1 to 5, stands out as the most fundamental one. The rating was determined by human annotators. Justification should be provided about the rating process , and how annotators agree with each other.  Supplement B4 and B5 do not include such information.

**Limitations:**

The evaluation process was carefully conducted, and covers a broader range of math problems. However, the evaluation highly depends on human experts. If evaluating another LLM on this set of math problems, human experts are required for making judgement on the LLM outputs.


**Opportunities For Improvement:**

1) The paper reported the evaluation primarily based on the miniGHOST dataset, which is a subset of GHOSTS. While this subset may provide valuable insights, it would be beneficial to include a more comprehensive datasets to ensure the generalizability.

2) This paper only uses the simple zero-shot prompt to evaluate LLMs. It would be interesting to see the few-shot performance, alongside some chain-of-thought approaches.


**Relation To Prior Work:**

Although the authors claims that the up-to-date development of AI in mathematical reasoning could be found in a survey. It is better to discuss some related work in the paper, such at chain-of-though approaches [1,2], benchmarks [3], etc.

[1] Chain-of-Thought Prompting Elicits Reasoning in Large Language Models
[2] Large Language Models are Zero-Shot Reasoners
[3] Lila: A Unified Benchmark for Mathematical Reasoning


**Summary And Contributions:**

The paper delves into the evaluate of ChatGPT and GPT-4 on mathematical tasks, aiming to evaluate their performance in question answering, theorem searching, and their potential to assist mathematicians. The paper contributes to the understanding of the mathematical capabilities of large language models and their potential applications.

---

> ### Author Response · Authors · 2023-08-10
> **Response to reviewer 3gvx (I)**
>
> Many thanks to reviewer 3gvx for his/her comments.  We answer below all the questions you have raised. We additionally would like to point out that we have uploaded a significantly revised version of our submission, that contains in total over 4 pages of further new material, including explanations and clarifications.
>
> > *The paper reported the evaluation primarily based on the miniGHOST dataset, which is a subset of GHOSTS. While this subset may provide valuable insights, it would be beneficial to include a more comprehensive datasets to ensure the generalizability.*
>
> The idea behind miniGHOSTS is that it's intentionally designed to be a smaller dataset, as opposed to something larger, in order to facilitate faster evaluation, since it is distilled from the GHOSTS dataset to capture a small dataset with heuristically similar performance.
>
> > *This paper only uses the simple zero-shot prompt to evaluate LLMs. It would be interesting to see the few-shot performance, alongside some chain-of-thought approaches.*
>
> We agree that investigating chain-of-thought [1] (or even more advanced methods, such as tree-of-thought [2]), which can also be mixed with various other forms of prompt engineering (e.g., majority voting), is an exciting undertaking. (A further important direction of study would be mathematical interaction traces, which model a more natural interaction of a user with a model, over an entire dialogue; this has been studied in [3].)
> At the same time, if we want to keep the current level of rigor, this would increase the evaluation methodology considerably, since, in addition to evaluating mathematics using the most complex evaluation protocol to date, which accounts for various types of failure modes using error and warning codes from Section B.4 (other datasets for mathematics, including the well-known MATH dataset, only use a binary correct-incorrect rating), one would have to account for a significant number of further failure modes, that can arise from documenting the models’ reasoning process.
>
> Because the current article is already quite long for a conference, exceeding 40 pages (including the supplementary material), we felt we would not be able to do justice to this important topic of using advanced in-context learning techniques by treating it in a cursory manner. Instead, we are preparing a separate publication, that focuses on these issues in much more depth, but on a smaller dataset that we have distilled out of GHOSTS, containing questions that, from GHOSTS, we know are hard for LLMs to answer. We aim to upload an arXiv preprint partially automating the rating and will try to reference that preprint in the camera-ready version of the current article.
>
> [1] J. Wei, Chain-of-Thought Prompting Elicits Reasoning in Large Language Models, https://arxiv.org/abs/2201.11903
> [2] S. Yao, D. Yuz, J. Zhao et al., Tree of Thoughts: Deliberate Problem Solving with Large Language Models, https://arxiv.org/abs/2305.10601
> [3] K. Collins, A. Jiang, S. Frieder et al., Evaluating Language Models for Mathematics through Interactions, https://arxiv.org/abs/2306.01694
>
> > *The evaluation process was carefully conducted and covered a broader range of math problems. However, the evaluation highly depends on human experts. If evaluating another LLM on this set of math problems, human experts are required to make judgments on the LLM outputs.*
>
> This can be partially automated, and we are working on it. As mentioned in the paper, our current solution to this is the miniGHOSTS approach, which distills the essential questions from GHOSTS. This is a novel approach to easing the evaluation burden on subsequent LLMs. We are in the process of turning this approach into a more systematic investigation,

---

> > ### Author Response · Authors · 2023-08-10
> > **Response to reviewer 3gvx (II)**
> >
> >
> > > *The shared datasets have no ‘ground-truth’ for problems, only the human evaluation on the GPT outputs. This makes it difficult for future usage. For the same problem, GPT may give different answers if running several times.*
> >
> > For a number of questions, in particular, those where the output should be a mathematical proof (which is perhaps the most interesting scenario, as it is also the most demanding, where the models received the lowest scores, see, e.g., the Olympiad-Problem-Solving subdataset of GHOSTS), it is not straightforward, what a ‘ground-truth’ could be: The essential point about having a ground-truth is that one can quickly compare the generated output to the ground truth.
> > But in the case of proofs, these can be presented in myriad ways: On one hand, for the same mathematical fact, one can find conceptually distinct proofs. (See [1] for an extreme example of this, where **122 conceptually distinct proofs** of the Pythagorean theorem are given.) On the other hand, one and the same proof can be presented in different ways, since the order in which the arguments are made and connected can be changed (e.g., if proving statements C depends on three statements X,Y,Z, we could prove first X,YZ and then conclude C; but we could also prove Y,Z,X and then conclude C - or construct the proof based on any other permutation of X,Y,Z).
> > Assuming all the previous problems were, somehow, solvable, further complicating the matters is the fact that not all possible proofs are known, and new ones are being discovered for known statements (staying with the topic of the Pythagorean theorem, even millennia after its publication, new proofs have surfaced [2]). If a model were to output a proof that is not known, it would still require a human to judge correctness, as in this case, there may not be any ground truth available to compare against.
> > Unfortunately, this makes it very hard for the ground-truth data to be included, and if it is included, it may not be meaningful for the outlined reasons. Some elementary datasets, such as the MATH dataset, have ground-truth data included in the comment. However, their ground truth is also susceptible to the issues we mentioned; human evaluation is necessary for absolute certainty.
> >
> > We thank the reviewer for raising this issue, and we will include the above explanations for why these issues do not apply in our paper to mitigate similar questions from readers.
> >
> > [1] Pythagorean Theorem https://www.cut-the-knot.org/pythagoras/
> > [2] B.F. Yanney , J. A. Calderhead, New and Old Proofs of the Pythagorean Theorem, https://www.tandfonline.com/doi/abs/10.1080/00029890.1896.11998759
> >
> > > *Another group of annotators may have different rating results.*
> >
> > Please consider Section B.5, where we have introduced mitigating measures against human errors.
> > Because annotators, in the case of mathematics, have to be expert which, by way of their education, underwent years of training in mathematics, we do not expect to see any systematic deviation in assessing whether a particular output of a model is correct or not (though there may be deviations among annotators in assessing how grave the annotator judge the mistake - as is typically the case when humans rate tests; the only solution, in this case, is to have multiple people rate the prompts, which, given limited human capacity, would have reduced the number of prompts we could evaluate; we trust on our broad set of mitigating measures in Section B.5 to reduce this type of human error as much as possible).
> >
> > > *Although the authors claims that the up-to-date development of AI in mathematical reasoning could be found in a survey. It is better to discuss some related work in the paper, such at chain-of-though approaches [1,2], benchmarks [3], etc.*
> >
> > Regarding [2], which advocates adding a “let’s think step-by-step,” we would like to point you to Section D.1, where we have tested this and found marginally better performance.
> >
> > Regarding [1], chain-of-thoughts (and, more generally, tree-of-thoughts), this investigation will be so comprehensive as to be better suited for a separate paper, as we argued above.
> >
> > We thank you for emphasizing these issues – we will update and emphasize these points more strongly in the paper, including referencing [3].

---

> > > ### Author Response · Authors · 2023-08-10
> > > **Response to reviewer 3gvx (III)**
> > >
> > > > *Reproducibility is unknown.*
> > >
> > > We are not quite sure what you mean - do you mean reproducibility in terms of arriving at the same evaluation or obtaining the same outputs out of (Chat)GPT?
> > > Originally we had included the dataset as a link to downloadable files here on OpenReview; since our revision, we have updated that, and the dataset is accessible now through our dataset website: https://ghosts.xyfrieder.xyz/
> > >
> > > In the first case, we note that, as with all humanly generated datasets, there will be a degree of variability, if humans attempt to recreate the dataset. This entails that almost all well-known datasets of a given complexity, such as the ImageNet ground truth, will have some degree of variability.
> > > We believe that reproducibility is something whose benefits are reserved for *experiments* rather than *datasets*, since for datasets, the concept of independent investigation of the validity of results by other researchers, the cornerstone of science, does not apply.
> > > This leads us to the second case: If you were referring to whether the outputs of ChatGPT on the prompts on our dataset can be reproduced, then unfortunately, it is beyond our control to ensure this. Even if API access is available and temperature is set to 0, there exist reports that the output is non-deterministic. [1,2]
> > > The fact that our analysis is made per-subdataset file, and not per-prompt, we are confident that a double-digit number of evaluations makes the conclusions that we draw robust: Even if, for a future evaluation, the model would output a wrong answer on some prompts, this would not skew the score over the overall subdataset files too much.
> > > We thank the reviewer for asking this question. We have added a section in the supplementary material in our recent update of our article (see Section "E Limitations and Reproducibility") that clarifies such reproducibility-related aspects in our paper.
> > >
> > > [1] S. Ouyang et al., LLM is Like a Box of Chocolates: the Non-determinism of ChatGPT in Code Generation, https://arxiv.org/pdf/2308.02828.pdf
> > > [2] S. Chann, Non-determinism in GPT-4 is caused by Sparse MoE, https://152334h.github.io/blog/non-determinism-in-gpt-4/
> > >
> > > > *The rating was determined by human annotators. Justification should be provided about the rating process , and how annotators agree with each other. Supplement B4 and B5 do not include such information.*
> > >
> > > We are unsure what the precise criticism is. We have added significantly more explanations in the mentioned sections in our latest update of the paper, and we hope that these will provide you with a clear picture (otherwise, please don't hesitate to ask us for further information).
> > > We would like to point out that we have also used these exact instructions from the mentioned sections among ourselves to enforce a uniform rating style successfully.

---

> > > > ### Author Response · Authors · 2023-08-30
> > > > **Response to reviewer 3gvx (IV)**
> > > >
> > > > We would like to thank you for taking the time to write the review and kindly ask if there is a way, to at least please acknowledge our rebuttal (that, we hope, covered all issues raised)? We have uploaded a significantly revised version of our submission, that contains in total over four pages of further new material, including extensive explanations and clarifications. If you have any further questions, please let us know.

---

> > > > > ### Comment · Reviewer_3gvx · 2023-08-30
> > > > >
> > > > > Thank you for the response and the update of the draft. We understand the intention behind using miniGHOSTS for faster evaluation and the considerations around maintaining the current level of rigor. However, considering that this paper targets on the investigation of the "Mathematical Capabilities of ChatGPT", we would still encourage you to explore the possibility of including additional datasets, even if smaller in scale, to enhance the generalizability of your findings. Regarding the few-shot evaluation, we understand that it might introduce complexities, but it could significantly enhance the practical applicability of your work and align it more closely with the real-world scenarios where ChatGPT could be employed. We understand your concern about increasing the evaluation methodology, but we believe that a balanced approach could potentially provide a more holistic understanding of ChatGPT's mathematical capabilities.
> > > > > We believe that a NeurIPS paper should have the strengthened impact by including comprehensive evaluation.

---

> > > > > > ### Author Response · Authors · 2023-08-30
> > > > > >
> > > > > > Dear reviewer, thank you for taking the time to get back to us. We appreciate that you have a busy schedule and are very happy you were able to respond.
> > > > > > The aim of our benchmark is to kickstart and facilitate future evaluation of language models using a novel methodology. As the field is constantly evolving, we tried to deliver a new type of benchmark; we are not claiming to prove that ChatGPT or GPT-4 are the state-of-the-art among all language models. Our evaluation was comprehensive, but not absolute (which may be a differentiating factor from NeurIPS Dataset&Benchmarks and NeurIPS main track).
> > > > > > Again, thank you for finding the time to answer, we appreciate the effort that went into your review.

---

### Official Review · Reviewer_2boP · 2023-07-20
**Good math-based analysis for ChatGPT**

**Rating:** 7
**Confidence:** 4
**Correctness:** 1. The paper seems generally correct …
**Clarity:** 1. The authors create new natural lan…

**Strengths:**

Here are four key strengths of this paper:

1. It provides a comprehensive evaluation of ChatGPT's mathematical capabilities across a wide range of domains, question types, and difficulty levels. The authors test ChatGPT on 6 custom-designed datasets as well as samples from existing datasets, covering topics from basic arithmetic to graduate-level proofs.

2. The methodology is rigorous, with clear rating criteria and a detailed taxonomy of error codes to analyze ChatGPT's failure modes. Each response is manually rated by experts in mathematics.

3. The paper introduces a new dataset called GHOSTS that focuses specifically on advanced mathematical comprehension, filling a gap compared to existing datasets that cover mainly elementary math. This dataset could become a valuable benchmark for testing future LLMs.

4. The analysis provides nuanced insights into ChatGPT's abilities and limitations in mathematics. It highlights cases where ChatGPT shows impressive performance as well as common mistakes and weaknesses.


**Additional Feedback:**

Please see comments in the sections above.

**Documentation:**

The paper investigates ChatGPT's mathematical capabilities by testing it on publicly available datasets as well as newly created ones and compares its performance to other models trained on mathematical data.

#### Main Contributions:

* Provides insight into ChatGPT's usefulness for mathematicians' workflows across different domains
* Identifies limitations and common failure modes of ChatGPT in advanced mathematics
* Introduces new natural language mathematics datasets (GHOSTS) for benchmarking ChatGPT and future LLMs

#### Key Findings:

* Contrary to positive media reports, ChatGPT struggles with graduate-level math exercises from textbooks and math competitions.
* It often makes mistakes in logical reasoning and computations.
* ChatGPT almost never expresses any form of uncertainty, even if its output has been completely wrong.
* However, it can be useful for searching and identifying mathematical concepts.

#### Key Recommendations:

* More natural language mathematics training data is needed for advanced reasoning.
* Incorporate capabilities for automatic evaluation and output verification.
* Enable expressing uncertainty and doubt in responses (chain-of-thought, tree-of-thought)


**Ethics:**

* The authors have made the full dataset publicly available for use by other researchers.

* The methodology seems sound and objective. The authors tested ChatGPT across a wide range of mathematical domains and difficulty levels. The rating system was carefully designed and well-documented.

* No deception was used in testing ChatGPT. The prompts were presented straightforwardly through the normal interface.

* The authors disclose that they have the required mathematical expertise to create and rate the dataset.


**Limitations:**

1. The datasets used to evaluate ChatGPT, while more advanced than previous benchmarks, still only cover a subset of undergraduate and beginning graduate-level mathematics. Truly testing the capabilities of ChatGPT on advanced graduate level or research mathematics would likely require even more difficult prompts.

2. The datasets are limited in size, with only 709 total prompts across 6 sub-datasets. Expanding the datasets to include more prompts and a wider diversity of mathematical topics would strengthen the evaluation.

3. As noted in the conclusion, the high cost of having mathematical experts manually rate each prompt response makes growing the datasets challenging for a single research group. More community participation is needed.

4. The evaluation is limited to certain types of mathematical reasoning (e.g. no geometry problems). Testing a broader range of mathematical skills could reveal other strengths or weaknesses..


**Opportunities For Improvement:**

There are also five opportunities for improvement in ChatGPT's mathematical capabilities:

1. The results show ChatGPT struggles with graduate-level mathematics, Olympiad problems, and complete proofs. Training on advanced mathematical corpora could help.

2. Strengthening logical reasoning. Flaws in logic and missing steps indicate issues with mathematical reasoning ability. More rigorously structured training with formal proofs may help in this area.

3. Expanding mathematical knowledge breadth. ChatGPT has gaps in coverage of advanced topics which limits capabilities. Continued pretraining on diverse mathematical texts and papers would expand knowledge.

4. Adding uncertainty awareness. Unlike other models, ChatGPT does not communicate a lack of confidence in mathematical answers. Uncertainty calibration would make the system more dependable.

5. Leveraging alternative modalities. As a text-based system, ChatGPT struggles with geometric problems. Incorporating diagrams and visual reasoning could expand skills.


**Relation To Prior Work:**

The paper discusses the related work on the mathematical capabilities of ChatGPT and other large language models in Section 2. Some key points from that section:

1. Automated mathematical reasoning has a long history dating back to 1959, with a focus on theorem proving. Recently there is more work on using machine learning and neural networks for mathematical reasoning.

2. Most recent large language models like PaLM and LaMDA are only evaluated on elementary math datasets like GSM8K, MathQA. The Minerva model stands out for being evaluated on the more advanced MATH dataset.

3. Existing evaluations of ChatGPT's math abilities consist of anecdotal evidence on the internet. This paper aims to do a more thorough investigation with a clearly defined methodology.

4. The MATH dataset used in this paper has unique features like requiring condensed numeric answers that allow automated evaluation. In contrast, this paper's datasets require expert ratings of free-form textual answers.


**Summary And Contributions:**

1. The authors show ChatGPT performs well on simple computational questions but struggles with advanced proofs and original problem-solving.

2. Identifies common failure modes of ChatGPT on mathematical reasoning, such as missing proof steps, incorrect computations, and faulty logic. This can help guide the future development of LLMs.

3. Introduces new natural language mathematics datasets (GHOSTS) curated by working mathematicians to benchmark capabilities on graduate-level concepts. This dataset is more advanced than existing ones.

4. Provides recommendations for using ChatGPT as an assistant for mathematicians, e.g. good for searching definitions but not reliable for proofs.

---

> ### Author Response · Authors · 2023-08-15
> **Response to reviewer 2boP (I)**
>
> Many thanks to reviewer 2boP for his/her comments. We would like to make a small remark regarding the statement *Introduces new natural language mathematics datasets (GHOSTS) curated by working mathematicians* from the **Summary and Contributions** section: We actually have released two datasets, GHOSTS, and miniGHOSTS, where the latter is a carefully distilled version of the former, in order to allow for more rapid evaluation.
>
> We go below into detail regarding various issues that were raised in the **Opportunities for Improvement**, **Limitations**, and **Documentation** sections:
>
> > *The results show ChatGPT struggles with graduate-level mathematics, Olympiad problems, and complete proofs. Training on advanced mathematical corpora could help.*
>
> We agree and would be very keen to do so. Unfortunately, ChatGPT and GPT-4 are closed-source, so we cannot influence their training in any way. Unfortunately, fine-tuning for these models via their API is not yet available at the time of writing [1]. We are in the process of fine-tuning other models on our data.
>
> [1] https://platform.openai.com/docs/guides/fine-tuning
>
> > *Strengthening logical reasoning. Flaws in logic and missing steps indicate issues with mathematical reasoning ability. More rigorously structured training with formal proofs may help in this area.*
>
> The various error codes of type "e5" are reserved to flag various flaws in logical reasoning (in particular, error code "e5_2" is reserved to flag missing steps), see section B.4.
> We agree that structured training with formal proofs may help, but training on and evaluating formal proofs is outside the scope of the current investigation, since natural proof-formal proof interfaces bring a whole host of other issues with them, please see [2].
>
> [2] Z. Azerbayev et al., ProofNet: Autoformalizing and Formally Proving Undergraduate-Level Mathematics, https://arxiv.org/abs/2302.12433
>
> > *Expanding mathematical knowledge breadth. ChatGPT has gaps in coverage of advanced topics which limits capabilities. Continued pretraining on diverse mathematical texts and papers would expand knowledge.*
>
> Earlier models, such as GPT-J [3], were trained on "The Pile" dataset [4], which contains a significant amount of mathematics. In particular, it contains as data the contents from arXiv, Wikipedia, as well as the Stack Exchange data dump (all of which contain a significant amount of mathematics). Nonetheless, it is known that these models do not outperform ChatGPT or GPT-4, which were released later, and, in all likelihood, were also trained on this data (exact training methodology is not; though for GPT-3 it is known that its training data includes Wikipedia [5]).
> Hence we believe that the issue is not that the mathematics is unknown to the models we tested (since it has probably "seen" during its pertaining stage more mathematics than any human has), but rather that it cannot turn all the mathematics it saw during training into working/operational knowledge to answer the questions we asked it correctly (as GHOSTS demonstrates, see Figures 1 and 5). To make the reasoning capabilities of the models better, approaches like "process supervision" [6] seem to be promising.
>
> [3] B. Wang, A. Komatsuzaki, GPT-J-6B: A 6 Billion Parameter Autoregressive Language Model., https://github.com/kingoflolz/mesh-transformer-jax
> [4] L. Gao et al., The Pile: An 800GB Dataset of Diverse Text for Language Modeling, https://arxiv.org/pdf/2101.00027.pdf
> [5] T.B. Brown, Language Models are Few-Shot Learners, https://arxiv.org/pdf/2005.14165.pdf
> [6] H. Lightman et al., Let's Verify Step by Step, https://arxiv.org/abs/2305.20050
>
> > *Adding uncertainty awareness. Unlike other models, ChatGPT does not communicate a lack of confidence in mathematical answers. Uncertainty calibration would make the system more dependable. Leveraging alternative modalities. As a text-based system, ChatGPT struggles with geometric problems. Incorporating diagrams and visual reasoning could expand skills.*
>
> We appreciate the suggestions. We have deliberately excluded these additional features (uncertainty, visual reasoning) because they make it hard to compare models.
> A cornerstone of our paper is the diagram comparing the performance of successive versions of ChatGPT, see Figure 2 (in addition to comparisons to other machine learning models that we made in-text); a fair comparison is only possible if all models support the same modalities.
> Currently, GPT-4 is the only significant model to support visual reasoning (via plugins); therefore, it was necessary to default to text-based input for a fair comparison.
> (Adding uncertainty in a principled way would have been interesting, but unfortunately requires access to the model's internals).
> We thank the reviewer for this remark and will incorporate it in the paper in order to clarify why we only considered text-based inputs.

---

> > ### Author Response · Authors · 2023-08-15
> > **Response to reviewer 2boP (II)**
> >
> >
> > > *The datasets used to evaluate ChatGPT, while more advanced than previous benchmarks, still only cover a subset of undergraduate and beginning graduate-level mathematics. Truly testing the capabilities of ChatGPT on advanced graduate level or research mathematics would likely require even more difficult prompts.*
> >
> > Since even GPT-4 was already struggling with graduate-level math (chapter 2 from Rudin), we felt that using even more difficult prompts would not have provided particular insight (other than the model not being able to solve them).
> >
> > > *The datasets are limited in size, with only 709 total prompts across six sub-datasets. Expanding the datasets to include more prompts and a wider diversity of mathematical topics would strengthen the evaluation.*
> >
> > We are working on this. Please note that while there are 709 prompts, the full dataset also consists of the evaluations, which was the harder part than devising/collecting the prompts. In total, 1636 evaluations have been carried out.
> >
> > > *As noted in the conclusion, the high cost of having mathematical experts manually rate each prompt response makes growing the datasets challenging for a single research group. More community participation is needed.*
> >
> > In the datasheet attached to the end of this paper, we have explicitly encouraged community participation, please see Question 58 and Question 55.
> >
> > > *The evaluation is limited to certain types of mathematical reasoning (e.g. no geometry problems). Testing a broader range of mathematical skills could reveal other strengths or weaknesses.*
> >
> > While geometry problems do not form their own subdataset, a significant number of problems are geometry problems. These are spread throughout various files and subdataset (e.g., MATH). To identify which problems are geometry problems, we have tagged each mathematical query with a Math Subject Classification code (https://zbmath.org/static/msc2020.pdf). Using these codes, it is straightforward to identify from our dataset which problems are geometry problems - or other types of problems.
> > We invite the reviewer to consider Figure 4 (supplementary material), where we displayed all important MSC codes and their ratings, as they show various domains of mathematics that are cross-spread through our dataset. In particular, it can be observed that on geometry problems, ChatGPT (version 9-January-2023) scores just below a passing grade of 3.5.
> >
> > > *Incorporate capabilities for automatic evaluation and output verification.
> >
> > We have investigated automatic evaluation capabilities, and it will result in a separate publication since there are many ways to (partially) automate such an investigation. Because the current article (including supplementary material) already exceeds 40 pages and the main body of the paper is densely packed with information related to the current study, we believe there simply would not have been enough space to do justice to a detailed investigation regarding automatic evaluation capabilities.
> >
> > ---
> >
> > If our clarifications --as well as our improvements to the paper based on your suggestions-- addressed your concerns, we would appreciate it if you could amend your score.
> > Conversely, if you have any other questions, please let us know; we will be very happy to answer them.
> >
> > ---

---

> ### Comment · Reviewer_2boP · 2023-08-30
> **Thank you for addressing my questions!**
>
> I carefully considered your clarification on those points and the insights shed new light on the matter. The automatic evaluation should be put in the future work section, it is very helpful for other researchers.

---

### Official Review · Reviewer_Sgwr · 2023-07-21
**Clear task, Detailed experiments and analysis, Detailed documentation, Limited innovation**

**Rating:** 7
**Confidence:** 4
**Correctness:** Yes
**Clarity:** Yes

**Strengths:**

1. The paper introduces a new methodology to evaluate the mathematical capabilities of large language models.
2. The creation of the GHOSTS dataset, covering graduate-level mathematics and distinguishing different dimensions of mathematical reasoning.
3. Evaluation of ChatGPT and GPT-4 on various mathematical tasks, providing insights into their strengths and weaknesses.
4. By emulating use cases that mathematicians encounter in their daily professional activities, the paper assesses the practical applicability of ChatGPT and GPT-4 as mathematical assistants.

**Additional Feedback:**

1. Many questions in evaluation require strong professional knowledge (such as Olympic competition questions). How to control the professionalism of annotators during human evaluation?

**Documentation:**

Yes

**Opportunities For Improvement:**

1. Mathematical problems usually require long-distance reasoning, so it can be considered to increase the evaluation of intermediate processes [1].
2. Most of the tasks involved in this work are biased towards mathematical proof, while the evaluation of algebraic operations is relatively small. Considering that algebraic operations are also an important part of errors in current mathematical problem-solving, it is possible to distinguish between operational errors and inference errors. [2]

[1] Lightman, Hunter, et al. "Let's Verify Step by Step." arXiv preprint arXiv:2305.20050 (2023).
[2] Yuan, Zheng, et al. "How well do Large Language Models perform in Arithmetic tasks?." arXiv preprint arXiv:2304.02015 (2023).

**Relation To Prior Work:**

Yes

**Summary And Contributions:**

The paper investigates the ChatGPT and GPT-4's mathematical capabilities using a novel methodology by a new dataset called GHOSTS.
They benchmark ChatGPT on GHOSTS and GPT-4 on a subset of GHOSTS and evaluate various performance metrics and show promising improvements in performance. They also acknowledges the flexibility of (Chat)GPT as a universal tool suitable for any area of mathematics, particularly in searching for mathematical objects. The authors express the need for further evaluation and encourage contributions to the dataset to establish a benchmark for assessing the mathematical abilities of language models.

---

> ### Author Response · Authors · 2023-08-12
> **Response to reviewer Sgwr**
>
> Many thanks to reviewer 9g86 for his/her comments. We go below into detail regarding two issues that were raised:
>
> > *Mathematical problems usually require long-distance reasoning, so it can be considered to increase the evaluation of intermediate processes [1].
> [1] Lightman, Hunter, et al. "Let's Verify Step by Step." arXiv preprint arXiv:2305.20050 (2023). *
>
> We agree, process supervision, as you indicated in [1], looks very promising. Unfortunately, we would need access to ChatGPT / GPT-4 internals to carry it out, beyond the ability to fine-tune the model [2], which is currently not yet enabled for GPT-4.
> Indeed, we look forward to other researchers using our dataset to carry out process supervision for their own needs, similar to how [1] used the MATH dataset. We expect that our approach of using richly annotated mathematical data (beyond binary correct-incorrect, as in the case of MATH) will benefit process supervision.
> We thank the reviewer for this observation and will include our remark above in the paper.
>
> [2] https://platform.openai.com/docs/guides/fine-tuning
>
> > *Most of the tasks involved in this work are biased towards mathematical proof, while the evaluation of algebraic operations is relatively small.*
>
> The Symbolic-Integration subdataset implicitly uses arithmetic reasoning, since in order to solve the integration problems in that dataset, one needs to carry out arithmetic computations in the course of, e.g., partial integration. Implicit arithmetic reasoning also occurs in the MATH subdataset, when solving math word problems (although we, of course, have no knowledge whether internally the model is indeed carrying out something resembling, say, partial integration, or is simply matching patterns; we suspect it is the latter since often the constants are wrong when it comes to integration).
> We have chosen to include a significant number of mathematical proofs in GHOSTS (where some of the proofs need snippets of arithmetic reasoning), rather than pure arithmetic reasoning, since:
> - our internal investigations indicated that these LLMs were not very proficient in terms of mathematical reasoning (and, in our view, this not be a suitable task to pose to a pure LLM, but instead would be a task for augmented models, such as Toolformers that contain dedicated arithmetic reasoning capabilities [3])
> - previous work partially covered arithmetic reasoning [4]
> - as mentioned, some of our subdatasets implicitly include arithmetic reasoning.
>
> We will include this remark in the paper to be more clear about our reasons for setting up the dataset the way we did.
>
> [3] T. Schick et al., Toolformer: Language Models Can Teach Themselves to Use Tools, https://arxiv.org/abs/2302.04761
> [4] D. Saxton et al., Analysing Mathematical Reasoning Abilities of Neural Models, https://arxiv.org/abs/1904.01557
>
> > *Considering that algebraic operations are also an important part of errors in current mathematical problem-solving, it is possible to distinguish between operational errors and inference errors. [2]
> [2] Yuan, Zheng, et al. "How well do Large Language Models perform in Arithmetic tasks?." arXiv preprint arXiv:2304.02015 (2023)*
>
> Error code "e4" (please see section B.4) is explicitly designed to distinguish operational errors, for all the other types of errors (inferences errors, logical errors etc., that are encoded in the various other error codes.
> We will update the paper to make the scope of what error code e4 pertains to more clear.

---

### Official Review · Reviewer_7RM4 · 2023-07-21
**Review for Submission 205**

**Rating:** 5
**Confidence:** 4
**Correctness:** Please refer to the "Opportunities Fo…

**Strengths:**

1. The paper introduces a new dataset, GHOSTS, that can be used to evaluate the mathematical performance of the existing LLMs.
2. The paper studies the math capability of ChatGPT and GPT-4 with the new dataset, and concludes that ChatGPT is good at understanding the question, but is not good enough for answering.
3. The authors also show that the GPT has been improved on math since the first release, and what the latest GPT-4 improves.

**Additional Feedback:**

NA

**Clarity:**

The paper is well written with missing details of the dataset construction and evaluation.

**Documentation:**

NA

**Limitations:**

The evaluation is limited to the models in ChatGPT family, while there are many open-source/commercial models that exist. Without the evaluation with those models, the new dataset/benchmark becomes a bit weak, as it is unclear how good it is to distinguish the models.

**Opportunities For Improvement:**

1. The source of the dataset is not well motivated. For example, why only the first two chapters of the Grad-Text are chosen instead of randomly sample problems from the whole textbook, while in Olypiad-Problem-Solving all the chapters are used to select the problems.
2. The paper lacks some insights of why the different subdatasets are needed and how that can help distinguish the existing models. It would be better to explicitly show some analysis about what one model is good at but another is not. It would be also good to show that what makes the original MATH dataset not sufficient for the evaluation, i.e., what is missing there with intuitive examples.
3. The evaluation does not include details, such as how the human ratings are done: is it crowd-sourcing or just a single author label them manually? What is the criteria used to assign the rating: does a wrong final result lead to a score 0 or the intermediate process lead to the result also count, which will further lead to questions, like if the intermediate process is considered for the rating, does that mean we are biased to model that tends to output longer answer?

**Relation To Prior Work:**

NA

**Summary And Contributions:**

The paper introduces a new dataset GHOSTS, which examines the mathematical capability of language models. With the different aspects of the math problems in GHOSTS, the authors examine the ability of ChatGPT and GPT-4 to solve math problems.

---

> ### Author Response · Authors · 2023-08-10
> **Response to reviewer 7RM4 (I)**
>
> Many thanks to reviewer 7RM4 for his/her comments. Before we go into detail regarding the various issues raised, we would like to clarify some potential misunderstandings that we believe might have come up.
>
> **Potential misunderstandings:**
>
> > *The paper introduces a new dataset, GHOSTS, that can be used to evaluate the mathematical performance of the existing LLMs.*
>
> We actually introduce two datasets, GHOSTS, and miniGHOSTS, where the latter was devised using the former using procedures detailed in the paper. The advantage of miniGHOSTS is that it allows to speed up evaluation, by asking language models only "representative" questions (which questions are representative is heuristically determined using GHOSTS).
>
> > *The paper studies the math capability of ChatGPT and GPT-4 with the new dataset, and concludes that ChatGPT is good at understanding the question, but is not good enough for answering.*
>
> We further have concluded (Section 5) that ChatGPT performs very well as a mathematical search engine, that its output is not consistent, and that for certain domains or question types of mathematics, its answer is sufficiently good to receive a passing grade.
>
> **Issues and problems that were raised by the reviewer:**
>
> > *The source of the dataset is not well motivated. For example, why only the first two chapters of the Grad-Text are chosen instead of randomly sample problems from the whole textbook, while in Olypiad-Problem-Solving all the chapters are used to select the problems.*
>
> There is a specific reason for this: The books in Grad-Text are textbooks, and as such, the level of difficulty and mathematical sophistication typically increases from chapter to chapter, in particular at the beginning of the books, as the foundations are established (e.g., the first chapter of the book from Rudin contains easier exercises than the second chapter; this is reflected in Figure 1 from our paper). Since the evaluated models fared poorly on the questions from the second chapter, it would not have made any sense to try even questions from later chapters, which are even harder.
> On the other hand, chapters from the book used for Olympiad-Problem-Solving are disconnected from each other, as each chapter revolves around a different problem-solving method, where one method is neither easier, nor harder than the other. Therefore, sampling randomly from that book preserves difficulty.
> We will include a short remark on this in the article to exclude any potential confusion.
>
> > *The paper lacks some insights of why the different subdatasets are needed and how that can help distinguish the existing models.*
>
> The selection of subdatasets was made in such a way that three core attributes/tags (Mathematical difficulty, Question type, and Types of high out-of-distribution likelihood) were covered on a number of values each attribute/tag can take (see Table 1). The subdatasets cover a large number of possible combinations of attribute values; the models' performance therefore provides answers to questions of general interest such as "for mathematically easy questions, where the model is asked to complete proofs, how well that (Chat)GPT perform?" Such questions are represented by tags M1 and Q5, and, e.g., the *Proofs Collection B Precalculus* subdataset from Table 1 matches it.  Figure 1 then shows the models' performance on this subdataset.
>
> > *It would be better to explicitly show some analysis about what one model is good at but another is not.*
>
> Please see Figure 1, where all evaluated models are rated against each other on each subdataset.
>
> > *It would be also good to show that what makes the original MATH dataset not sufficient for the evaluation, i.e., what is missing there with intuitive examples.*
>
> As Table 1 shows, the original MATH dataset only covers a few values of the possible attributes, which made it necessary to add more subdatasets to explore the full range of values.
>
> > *The evaluation does not include details, such as how the human ratings are done: is it crowd-sourcing or just a single author label them manually? What is the criteria used to assign the rating: does a wrong final result lead to a score 0 or the intermediate process lead to the result also count, which will further lead to questions, like if the intermediate process is considered for the rating, does that mean we are biased to model that tends to output longer answer?*
>
> Please see the supplementary material, Sections B.4 and B.6, that fully addresses these questions. We have also answered the questions regarding dataset authorship in the datasheet at the end of the paper.

---

> > ### Author Response · Authors · 2023-08-10
> > **Response to reviewer 7RM4 (II)**
> >
> > > *The evaluation is limited to the models in ChatGPT family, while there are many open-source/commercial models that exist. Without the evaluation with those models, the new dataset/benchmark becomes a bit weak, as it is unclear how good it is to distinguish the models.*
> >
> > ChatGPT and GPT-4 dominates all existing open-source model [1, 2], so we chose to focus on the most promising models for the challenging dataset we devised.
> >
> > [1] OpenAI, GPT-4 Technical Report, https://arxiv.org/abs/2303.08774
> > [2] S. Bubeck et al., Sparks of Artificial General Intelligence: Early experiments with GPT-4, https://arxiv.org/abs/2303.12712

---

> > > ### Comment · Reviewer_7RM4 · 2023-08-16
> > > **Re: Response to reviewer 7RM4**
> > >
> > > Thank you for the detailed reply!
> > >
> > > > It would be better to explicitly show some analysis about what one model is good at but another is not.
> > > >> Please see Figure 1, where all evaluated models are rated against each other on each subdataset.
> > >
> > > Thanks for the pointer! I saw the figure for the results, but I was expecting more insights/analysis regarding whether one category is more correlated to another, and if a model is good on one category, does that mean it is generalizable to the other categories?
> > >
> > > > ChatGPT and GPT-4 dominates all existing open-source model [1, 2], so we chose to focus on the most promising models for the challenging dataset we devised.
> > >
> > > Thanks for the reply! Yes, the ChatGPT and GPT-4 are widely known dominates all existing open-source model in existing benchmarks, but it is always good to understand if the newly proposed benchmark aligns with the previous ones, or it provides any more insights of the performance comparison for the existing models, e.g., if a model specialized on code will work better for some math problems.
> > >
> > > > The evaluation does not include details, such as how the human ratings are done: is it crowd-sourcing or just a single author label them manually? What is the criteria used to assign the rating: does a wrong final result lead to a score 0 or the intermediate process lead to the result also count, which will further lead to questions, like if the intermediate process is considered for the rating, does that mean we are biased to model that tends to output longer answer?
> > > >> Please see the supplementary material, Sections B.4 and B.6, that fully addresses these questions. We have also answered the questions regarding dataset authorship in the datasheet at the end of the paper.
> > >
> > > Thanks for the pointer to the details, which partially resolve my concern. It would be nice to include the pointer in the main paper as well.
> > > I still have the following concerns about the evaluation process:
> > > 1. Although B.6 mentions that the evaluations are done by a subset of the authors, it would be nice to show the actual number of people involved for each question.
> > > 2. Additionally, is the rating consistent across multiple authors? What does the variance look like? How easy it would be for a following up paper that adopts the benchmark in the paper to replicate the result if they do not have the same set of authors in their paper?
> > >
> > > ---
> > > After reading the review comments and the authors' reply, I think the dataset itself might be good for the community and involves a lot of effort, but I am fully convinced by the selection of the categories and the evaluation. Therefore, I am going to increase my score to 5.

---

> > > > ### Author Response · Authors · 2023-08-28
> > > >
> > > > We thank reviewer 7RM4 for his update and for taking the time to follow up with our rebuttal, for his/her follow-up questions, which helped elucidate some aspects of our evaluation, as well as his score increase.
> > > >
> > > > We will below answer the remaining questions (and are happy to answer subsequent ones, should ones ones arise):
> > > >
> > > > > *I was expecting more insights/analysis regarding whether one category is more correlated to another, and if a model is good on one category, does that mean it is generalizable to the other categories?*
> > > >
> > > > We agree that comparisons across mathematical problem categories and how performance correlates (or, more generally, what the statistical dependency relation is - correlation is just a measure of linear relationship) is indeed a very important issue.
> > > >
> > > > But it is slightly unclear, how to interpret "correlation". In its strictest sense, in order to test correlation (https://en.wikipedia.org/wiki/Correlation), repeated experiments are necessary on which two different random variables based, and correlation/statistical dependency can then be measured for these.
> > > > A setting that naturally fits into this framework would be one where we finetune the (Chat)GPT models on one type of questions (e.g., geometry questions), and then evaluate on questions of another type (e.g. number theory questions). Unfortunately, is not possible, since we do not have the code.
> > > > (A subset of the authors is in the process of preparing a systematic evaluation, that aims to address this question; since this work will be comprehensive, and the current article has already exceeded 40 pages (including supplementary material), we have decided to make a separate publication regarding this endeavor.)
> > > >
> > > > Therefore, we believe what you could have in mind is perhaps performing an ablation study, where a model is fine-tuned only on one type of problems?
> > > >
> > > > (If we are interested in correlations among mathematical domains, where we only have a fixed model, we believe the figures show the correlations between different datasets well.)
> > > >
> > > > > *Although B.6 mentions that the evaluations are done by a subset of the authors, it would be nice to show the actual number of people involved for each question.*
> > > >
> > > > Five of the authors of the study were tasked to evaluate the prompt in an initial phase. Their evaluation was subsequently checked for various types of errors and the dataset was streamlined (see section B.5, "Mitigating Human Errors").
> > > >
> > > > > *Additionally, is the rating consistent across multiple authors? What does the variance look like? How easy it would be for a following up paper that adopts the benchmark in the paper to replicate the result if they do not have the same set of authors in their paper?*
> > > >
> > > > On the level of files that make up the subdatasets (see Fig. 1 for the files in each subdataset) no two authors worked on the same file (and only the MATH and Holes-in-Proofs subdataset had files on which multiple authors worked; for the other subdatasets a single authir evaluated the entire subdataset).
> > > > Therefore, because each file treats a different type of mathematics, comparisons of rating would not be fair, since authors' ratings should be compared on the same type of questions. Nonetheless on the checks that we performed  (see section B.5, "Mitigating Human Errors"), we did not observe significant rating divergence. We, therefore, believe that a re-evaluation of the output would at best on the level of subdatasets show only a small shift in scores compared to the ones we obtained.

---

### Official Review · Reviewer_9g86 · 2023-07-24
**The paper introduces a new natural language mathematics dataset called GHOSTS to benchmark the mathematical capabilities of large language models (LLMs) like ChatGPT and GPT-4.**

**Rating:** 8
**Confidence:** 3

**Strengths:**

- It introduces a new natural language mathematics dataset, GHOSTS, to test language model capabilities on advanced mathematical comprehension. This helps fill a gap, as most existing datasets cover only elementary mathematics.

- The dataset is carefully designed to cover different aspects of mathematical reasoning like stating facts, computational questions, proof completions, and solving Olympiad-style problems. This provides a more comprehensive test of language models.

- The authors perform a rigorous analysis of ChatGPT and GPT-4 using their proposed methodology of ratings, error codes, and warnings. This sheds light on the models' mathematical abilities and limitations.

- Benchmarking on their dataset shows ChatGPT and GPT-4 lag behind specialized models trained on math tasks. But GPT-4 shows promising improvements. This helps track progress.

- The work identifies types of questions and workflows where ChatGPT could assist mathematicians versus where its capabilities are limited. This guides proper integration into mathematicians' workflows.

- The dataset required extensive human effort and mathematical expertise to create. This establishes a thorough benchmark for evaluating language models on math.

- The authors encourage extending the dataset and evaluation to drive progress on language models for mathematics.

**Additional Feedback:**

Please see the comments above.

**Clarity:**

Overall, the paper is well-written and clearly presents the key information:

- The introduction provides helpful context and clearly motivates the need for the work. The objectives are well-defined.

- The authors provide extensive details on the dataset composition and methodology to support reproducibility.

- The results are presented logically using visualizations, statistics, and examples. The performance analysis flows well.

- The conclusion concisely summarizes the main findings and implications.

- The writing is clear and concise throughout. The technical explanations avoid ambiguity.

- The paper is well-structured, making it easy to follow. Related concepts are grouped logically.

Some areas that could further improve clarity:

- The abstract could provide more concrete details about the key findings rather than just a high-level summary.

- Additional visualizations and examples demonstrating model capabilities could further ground the concepts.


**Correctness:**

- The methodology for prompting the models and evaluating their outputs seems sound. The rating scheme and error codes are well-defined.

- The authors utilize two existing datasets (MATH and Symbolic Integration) with known providence as part of GHOSTS. This supports the benchmark's integrity.

- The analyses seem to be performed correctly without obvious flaws. The statistics and visualizations appear accurate based on the reported methods.

- The authors seem open about the limitations and do not overstate claims beyond what the results support.

- The findings align with expectations about the models' mathematical reasoning abilities based on their training objectives.


**Documentation:**

The authors provide a comprehensive datasheet in the appendix that covers important details about the dataset:

- It describes the motivation, composition, creation process, intended uses, and other key information about each subdataset.

- It provides statistics on the dataset composition including size, question types, mathematical difficulty, etc.

- The authors describe the extensive human effort required for creation and annotation.

- Licensing information allows public access and reuse of the dataset.

- Maintenance plans are discussed - the authors encourage community contributions.

- Ethical considerations around potential misuse are acknowledged.

Additionally:

- The methodology for prompting and evaluating models is clearly documented to support reproducibility.

- The authors provide some examples of dataset samples and model outputs.

- Code and detailed instructions for reproducing the benchmark do not seem to be included. The authors could consider providing these assets.

Overall, the dataset is well-documented, and key information is provided to support reuse in derivative works. Some additional implementation details could further improve reproducibility of the benchmark.

**Limitations:**

 The authors have made a good effort to discuss the limitations and potential negative impacts of their work, but there are still some areas that could be expanded on:

- The dataset creation process required extensive human effort and mathematical expertise. The limitations around scaling up the dataset size are acknowledged, but more discussion could be had around potential solutions or alternative approaches.

- The benchmarking is currently limited to certain models and tasks. Expanding the scope would yield more comprehensive results. The authors recognize this, but more detail on specific limitations of the current benchmark would be helpful.

- There is limited discussion of potential misuse cases and steps to mitigate harm. For example, ChatGPT could produce erroneous mathematical solutions that mislead students. The authors could suggest best practices for use.

- More analysis of why models fail certain tasks through ablation studies or error analysis could be suggested as future work.

- Comparisons to human performance are lacking. Contrasting with professional mathematicians could better contextualize the capabilities.

Overall, the authors have made a solid effort to acknowledge limitations and potential negative impacts. But the work would be strengthened by expanding the discussion on these topics and providing more constructive suggestions to address them in future research.

**Opportunities For Improvement:**

- The GHOSTS dataset, while more comprehensive than others, is still limited in size and coverage. Expanding the dataset with more domains, question types, and levels of difficulty could better measure model capabilities.

- The benchmarking is currently only on two versions of ChatGPT and an early version of GPT-4. Evaluating more models, especially newer releases, would give a clearer picture of progress over time.

- While the analysis identifies strengths and weaknesses of the models, it does not diagnose the underlying reasons behind their performance. Further probing could uncover strengths/weaknesses in the models' reasoning.

- Ablation studies and error analyses may provide more insight into why models fail certain tasks. This could drive architecture improvements.

Overall, the work makes excellent progress but has opportunities to build an even more thorough mathematical benchmark and analysis. Expanding the dataset diversity and model evaluations would be the most impactful next steps.

**Relation To Prior Work:**

- The introduction clearly cites limitations of existing natural language mathematics datasets, motivating the need for GHOSTS.

- They provide a comprehensive overview of related works, comparing their approach to existing models and datasets.

- They highlight how GHOSTS covers more dimensions of mathematical reasoning versus prior datasets.

- They compare performance on specific subdatasets/tasks to specialized models, contextualizing ChatGPT and GPT-4's capabilities.

- They analyze the differences between ChatGPT versions and the improvements seen with GPT-4.

- They identify specific types of mathematical questions and workflows where ChatGPT excels or struggles compared to prior systems.

- They note the limitations of formal mathematics datasets based on symbolic encodings.

- The related work section situates the contributions among existing literature and clearly expounds the incremental advances.


**Summary And Contributions:**

The authors introduce GHOSTS, a new benchmark dataset for evaluating the mathematical reasoning capabilities of large language models (LLMs) like ChatGPT and GPT-4. The dataset comprises 709 prompts across 6 subdatasets covering a diverse range of mathematical skills, from basic arithmetic to advanced graduate-level concepts. A key contribution is the inclusion of carefully hand-crafted prompts to test abilities beyond existing math datasets.

The authors conduct a detailed analysis of two ChatGPT versions and GPT-4 using their proposed methodology. The key findings are:

Performance correlates with mathematical difficulty, with models struggling on complex proofs and original problems.
GPT-4 demonstrates substantially improved performance over ChatGPT.
ChatGPT shows promise as a mathematical search engine, but inconsistently produces high quality solutions.
They conclude ChatGPT is not yet suitable for rigorous exam-level work, contrasting hype about its math skills. The granular analysis amounts to a “model card” summarizing capabilities.

---

### Author Response · Authors · 2023-08-28
**Summary of all rebuttals**

Many thanks to all the reviewers. We are delighted to note the reviewers' affirmation that our dataset -beyond being a timely topic- fills a gap in existing literature. Further, we appreciate the recognition of the comprehensiveness of our dataset, its coverage of a diverse range of mathematical domains emulating use cases that mathematicians encounter, the rigorous nature of our benchmarking procedures, and of our detailed evaluation methodology.

We would like to draw attention to the reviewers that we uploaded a significantly updated version of our submission (which also supersedes the arXiv update that we have made since the original NeurIPS D&B submission). This update contains many improvements, and further explanations related to your inquiries: We have made full use of the additional page that could be added to the main text, and we have added three more pages worth of explanations and clarifications in the supplementary material. Our references list grew to a total of 62, and our total page size from 42 to 46.

We would also like to draw attention to several key aspects that may not have been fully considered by all reviewers:
- A cornerstone of our article is the comparative analysis between three iterations of ChatGPT models (version 9-January-2023, 30-January-2023, which, according to the release notes, has "improved mathematical abilities" and GPT-4), see Figure 2.
- We noticed that the large selection of advanced mathematical problems that we analyzed, which we display in Figure 4, and the other figures from the supplementary material, have not been commented on.
- The miniGHOSTS dataset is a novel type of dataset used for speeding up the evaluation, by heuristically containing the essential elements of GHOSTS, in order to speed up evaluation.

In light of these points, and in addition to various other improvements made to the manuscript, we kindly invite reviewers to please engage with our rebuttals: If, upon reading our updated submission, you find our revisions and additions to have significantly enhanced the submission's contributions, we would be most appreciative if you would consider revising your evaluation scores accordingly.

Once again, we thank you for your valuable input and look forward to your updated assessments.

---

### Decision · Program_Chairs · 2023-09-22

**Decision:**

Accept (Poster)

**Comment:**

Summary (key phrases borrowed from reviews)

The authors introduce GHOSTS, a new benchmark dataset for evaluating the mathematical reasoning capabilities of large language models like ChatGPT and GPT-4. The dataset comprises 709 prompts across 6 subdatasets covering a diverse range of mathematical skills, from basic arithmetic to advanced graduate-level concepts. A key contribution is the inclusion of carefully hand-crafted prompts to test abilities beyond existing math datasets.The dataset is curated by working mathematicians to benchmark capabilities on graduate-level concepts.

The authors then conduct a detailed analysis of two ChatGPT versions and GPT-4 on a novel methodology introduced by them. Performance correlates with mathematical difficulty, with models struggling on complex proofs and original problems. GPT-4 demonstrates substantially improved performance over ChatGPT. ChatGPT shows promise as a mathematical search engine, but inconsistently produces high quality solutions. They conclude that ChatGPT is not yet suitable for rigorous exam-level work, contrasting hype about its math skills.

Strengths

1. Most existing datasets cover only elementary mathematics, so this is a step in the right direction.
2. The dataset is carefully designed to cover different aspects of mathematical reasoning like stating facts, computational questions, proof completions, and solving Olympiad-style problems. This provides a more comprehensive test of language models.
3. The proposed methodology of ratings, error codes, and warnings is a novel new methodology to evaluate the mathematical capabilities of large language models
4. The work identifies types of questions and workflows where ChatGPT could assist mathematicians versus where its capabilities are limited. The dataset required extensive human effort and mathematical expertise to create. This establishes a thorough benchmark for evaluating language models on math. Also, Identifies common failure modes of ChatGPT on mathematical reasoning, such as missing proof steps, incorrect computations, and faulty logic. This can help guide the future development of LLMs.
5. Paper is clear, well written and sound.

Improvements

1. Can expand the dataset with more domains, question types, and levels of difficulty. The current number of 709 examples, while a good start, might be still very limited. This points to a difficulty of scaling the current approach of collecting examples, which might be mitigated by engaging the broader Mathematics community to build a larger dataset.
2. Comparisons to human performance are lacking. Contrasting with professional mathematicians could better contextualize the capabilities.
3. The paper only reports 0-shot performance and not few-shot and CoT performance, which are known to improve the mathematical capabilities of GPT-style models considerably. The authors respond that they aren't able to achieve this in this paper since they have committed to a rigorous evaluation, and the number of failure modes drastically increase with the addition of few-shot examples and CoT. They leave it to future work.